# *In vitro* study of Hesperetin and Hesperidin as inhibitors of zika and chikungunya virus proteases

Raphael J. Eberle[1,2☯]*, Danilo S. Olivier[3], Carolina C. Pacca[4,5], Clarita M. S. Avilla[4], Mauricio L. Nogueira[6], Marcos S. Amaral[7], Dieter Willbold[2,8,9], Raghuvir K. Arni[1], Monika A. Coronado[1,2☯]*

1 Multiuser Center for Biomolecular Innovation, Departament of Physics, Instituto de Biociências Letras e Ciências Exatas (Ibilce), Universidade Estadual Paulista (UNESP), São Jose do Rio Preto, SP, Brazil, 2 Institute of Biological Information Processing (IBI-7: Structural Biochemistry), Forschungszentrum Jülich, Jülich, Germany, 3 Federal University of Tocantins, Araguaína, TO, Brazil, 4 Instituto de Biociências Letras e Ciências Exatas (Ibilce), Universidade Estadual Paulista (UNESP), São Jose do Rio Preto, SP, Brazil, 5 FACERES Medical School, São José do Rio Preto, Brazil, 6 Faculdade de Medicina de São José do Rio Preto–FAMERP, São José do Rio Preto, Brazil, 7 Institute of Physics, Federal University of Mato Grosso do Sul, Campo Grande, MS, Brazil, 8 Institut für Physikalische Biologie, Heinrich-Heine-Universität Düsseldorf, Universitätsstraße, Düsseldorf, Germany, 9 JuStruct: Jülich Centre for Structural Biology, Forchungszentrum Jülich, Jülich, Germany

☯ These authors contributed equally to this work.
* r.eberle@fz-juelich.de (RJE); m.coronado@fz-juelich.de (MAC)

**Data Availability Statement:** All relevant data are within the paper and its Supporting Information files.

## Abstract

The potential outcome of flavivirus and alphavirus co-infections is worrisome due to the development of severe diseases. Hundreds of millions of people worldwide live under the risk of infections caused by viruses like chikungunya virus (CHIKV, genus *Alphavirus*), dengue virus (DENV, genus *Flavivirus*), and zika virus (ZIKV, genus *Flavivirus*). So far, neither any drug exists against the infection by a single virus, nor against co-infection. The results described in our study demonstrate the inhibitory potential of two flavonoids derived from citrus plants: Hesperetin (HST) against NS2B/NS3$^{pro}$ of ZIKV and nsP2$^{pro}$ of CHIKV and, Hesperidin (HSD) against nsP2$^{pro}$ of CHIKV. The flavonoids are noncompetitive inhibitors and the determined IC$_{50}$ values are in low µM range for HST against ZIKV NS2B/NS3$^{pro}$ (12.6 ± 1.3 µM) and against CHIKV nsP2$^{pro}$ (2.5 ± 0.4 µM). The IC$_{50}$ for HSD against CHIKV nsP2$^{pro}$ was 7.1 ± 1.1 µM. The calculated ligand efficiencies for HST were > 0.3, which reflect its potential to be used as a lead compound. Docking and molecular dynamics simulations display the effect of HST and HSD on the protease 3D models of CHIKV and ZIKV. Conformational changes after ligand binding and their effect on the substrate-binding pocket of the proteases were investigated. Additionally, MTT assays demonstrated a very low cytotoxicity of both the molecules. Based on our results, we assume that HST comprise a chemical structure that serves as a starting point molecule to develop a potent inhibitor to combat CHIKV and ZIKV co-infections by inhibiting the virus proteases.

**Funding:** The study was supported by the Conselho Nacional de Desenvolvimento Científico e Tecnológico (435913/2016-6, 401270/2014-9, 307338/2014-2) received by RA. Fundação de Amparo à Pesquisa do Estado de São Paulo (2016/12904-0, 2018/12659-0, 2018/07572-3, 2019/05614-3) by RE and MC. Ciência e Tecnologia do Estado de Mato Grosso do Sul (23/200.307/2014) by MA. The funders had no role in study design, data collection and analysis, decision to publish, or preparation of the manuscript.

**Competing interests:** The authors have declared that no competing interests exist.

## Introduction

Environmental, climatic and ecological changes create niches that support and stimulate the proliferation of arthropod-borne viruses in warmer developing countries [1]. Low investment in fragile health care systems and lack of potent antiviral drugs [2] make it difficult to control the spread of infectious diseases [2–4]. Due to these reasons, 390 million dengue virus (DENV) infections are reported annually and 128 countries are at risk of DENV infection [2, 5]. Recent epidemics due to chikungunya virus (CHIKV) and zika virus (ZIKV) affected millions of people in the Americas [6–9].

*Aedes aegypti* mosquitoes, which are prevalent over large geographical areas across different countries, are the prime vectors involved in the transmission of CHIKV (genus *Alphavirus*), DENV and ZIKV (genus *Flavivirus*) to humans. *A. aegypti* together with *A. albopictus*, also a more widely distributed mosquito, continue to emerge in new regions, due to climate change [10–12]. In fact, many regions in the Americas, e.g. Nicaragua, Costa Rica, Colombia, and Brazil, have recently experienced simultaneous outbreaks of CHIKV, DENV and ZIKV infections [6, 13, 14] (Fig 1A and 1B).

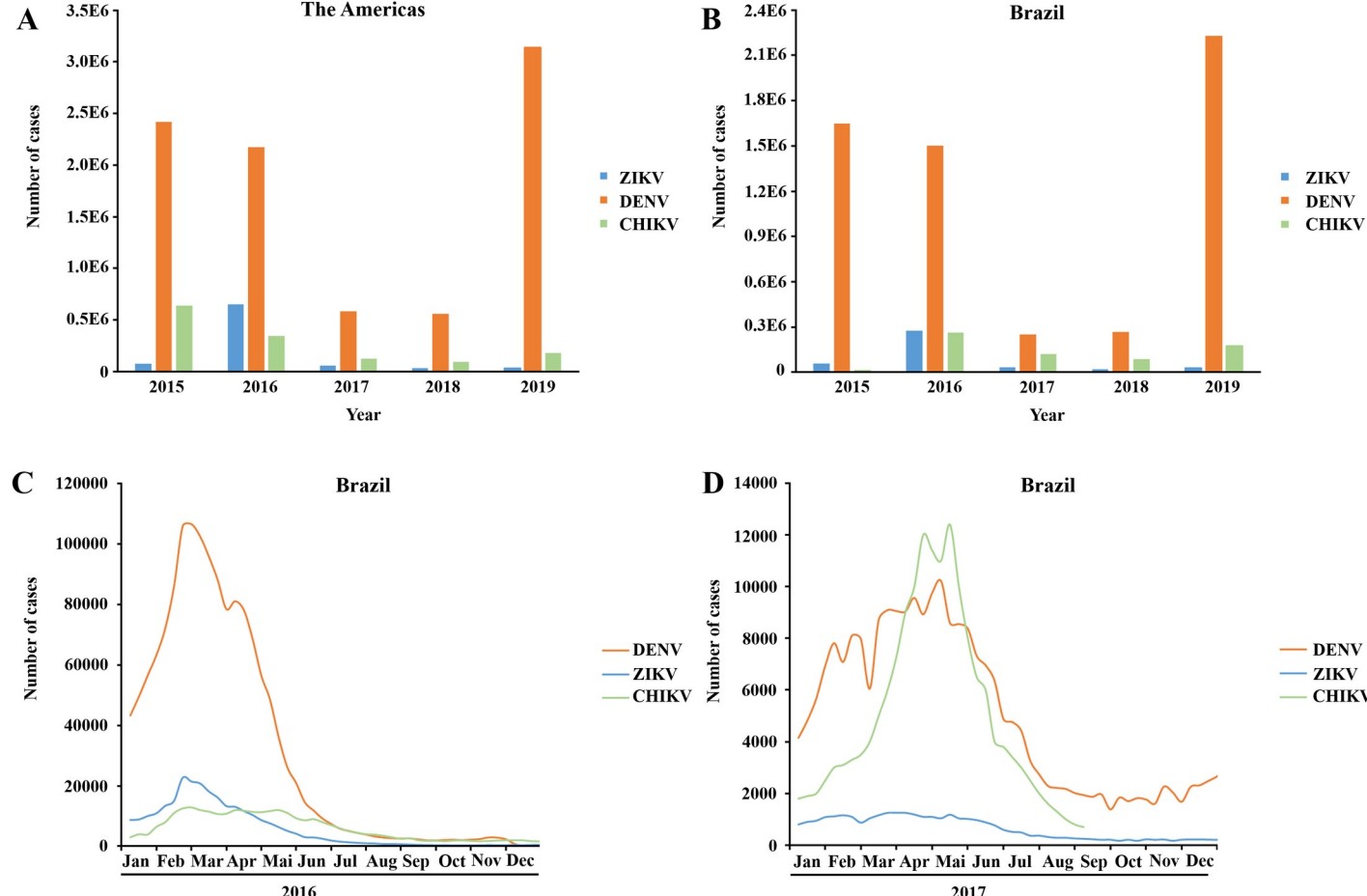

**Fig 1. Reported cumulative cases of CHIKV, DENV and ZIKV from 2015 to 2019 in the Americas and Brazil.** Based on PAHO and WHO (PAHO/WHO, 2020) [15]. ZIKV infections colored in blue, DENV in orange and CHIKV in green. **A:** Confirmed ZIKV, DENV and CHIKV cases in the Americas from 2015 to 2019. **B:** Confirmed ZIKV, DENV and CHIKV cases in Brazil from 2015 to 2019. **C:** Confirmed ZIKV, DENV and CHIKV cases in Brazil 2016. **D:** Confirmed ZIKV, DENV and CHIKV cases in Brazil 2017.

Simultaneous infections by two or all of these viruses have been reported [16–21]. Considering only the clinical manifestation, ZIKV, CHIKV and DENV infections can show similar clinical signs, making it difficult to identify the causative virus for the symptoms [22]. A recent study in Nicaragua demonstrated that 27% of the tested patients were positive for a co-infection with two viruses or all the three viruses described here [22].

The presence of CHIKV, DENV and ZIKV (Fig 1C and 1D) in Brazilian patients between 2016 and 2017 exhibited high incidences of simultaneous infections, and the co-infection caused by CHIKV, DENV, and ZIKV are alarming because of the development of secondary clinical conditions. ZIKV can be transmitted from a pregnant woman to her fetus that can cause a birth defect, for example, microcephaly [22]. Infections caused by CHIKV can induce chronic arthritis and cognitive disorders [8] together with indistinguishable febrile illness, including headaches, nausea, myalgia arthralgia, and rash [14]. Simultaneous transmission and co- infection of ZIKV and CHIKV is a concern in the Americas, because both the viruses co-circulate in the same geographical regions [16–21]. To control co-infections, it is vital to identify molecules with the potential to inhibit proteins that are essential for replication process of the viruses. Hence, the non-structural (NS) viral proteins are attractive targets for the development of drug-like molecules [23]. ZIKV contain seven NS proteins, NS1, NS2A, NS2B, NS3, NS4A, NS4B and NS5. The virus polyprotein is co- and post-translationally cleaved by the viral protease NS3 [24], which requires the NS2B cofactor for optimal enzymatic activity [25]. The NS2B/NS3, a serine protease (NS2B/NS3$^{pro}$), is essential for virus replication, making it an ideal target for drug development [26, 27]. CHIKV contain a cysteine protease (nsP2$^{pro}$), that have same vital functions as NS2B/NS3$^{pro}$ (ZIKV) in the virus replication process, which together with other non-structural proteins (nsP1, nsP3 and, nsP4), form the nonstructural polyprotein precursor [28].

Currently, neither any appropriate antiviral treatment for CHIKV and/or ZIKV infection is available, nor for treatment of co-infections caused by these viruses. Though, previous studies describe different classes of compounds with inhibitory potential against one of the aforementioned viruses, but none are available against both (S1 Table). Several of these compounds are botanical products [29–33], which are an attractive source in the search for novel drug candidates. Natural products form a large reservoir of bioactive compounds that can be considered as candidate for drug development [34]. Flavonoids, a huge group of polyphenolic compounds found ubiquitously in plants, exhibit a wide variety of biological effect such as anti-inflammatory, antioxidant, antiviral, antibacterial, cardiovascular and chemo preventive activities [35]. HSD (3',5,7-trihydroxy-4'-methoxy-flavanone-7-rhamnoglucoside) is a member of the flavanone group, a type of flavonoids, which is mainly isolated from citrus fruits. HSD is known to be partly deglycosylated in the human gut by the intestinal microbiota to its aglycone form HST (3',5,7-trihydroxy-4'-methoxyflavanone) [36]. Both flavonoids possess several pharmacological properties, e.g. antioxidant, anti-inflammatory and neuroprotective [37, 38]. Additionally, HSD and HST have been reported to inhibit intracellular replication of CHIKV [29]. Vander dos Santos and collaborators described inhibition of serine proteases by HST, isolated from snake venom [39], thereby demonstrating possible inhibitory potential of these molecules against different proteases.

Our study describes new aspects concerning the inhibitory effect of HST and HSD against the proteases (NS2B/NS3 and nsP2) involved in the replication process of ZIKV and CHIKV. HST inhibited both proteases with IC$_{50}$ values in the low μM range. HSD showed only a weak inhibitory effect against ZIKV NS2B/NS3$^{pro}$, which is in agreement with the results described by Lim et al. 2017 [40], however, it significantly inhibited nsP2$^{pro}$ of CHIKV. Activity tests in combination with molecular docking and MD simulations of HST and HSD in complex with the proteases, demonstrated a possible inhibition mode of the molecules. Furthermore, MTT

assays of both molecules (concentration of up to 300 μM) demonstrated a low cytotoxicity against Vero cells. The 50% cytotoxic concentration was not reached, and even if the plaque reduction assays showed no antiviral effect of HST and HSD against both viruses, our results demonstrate the potential of HST to serve as lead molecule against virus proteases.

## Material and methods

### Cloning, expression and purification of virus proteases

**ZIKV NS2B/NS3[pro].** ZIKV NS2B/NS3[pro] (GenBank Protein Accession number KU729217.2, Brazilian isolate BeH823339) was cloned, expressed and purified as described previously [41].

**CHIKV nsP2[pro].** The DNA fragment encoding CHIKV nsP2[pro] (residues 466–798) containing the N-terminal cysteine protease domain and the C-terminal SAM-dependent methyl-transferase domain (GenBank Protein Accession number AAN05101.1, strain S27-African prototype) was synthesized (BioCat GmbH, Heidelberg, Germany) and implemented in the kanamycin resistant vector pET-24a (+). The construct contained an N-terminal hexahistidine affinity tag and a TEV protease cleavage site (ENLYFQG).

*E.coli* (DE3) T1 strain (Sigma-Aldrich, USA) was transformed using pET-24a (+)-nsP2[pro] plasmid. LB medium supplemented with 50 μg/ml kanamycin was used for inoculation of *E. coli* (DE3) T1 containing pET-24a (+)-nsP2[pro]. The expression culture was grown at 37°C to an optical density of 0.6 at 600 nm ($OD_{600}$). The temperature was then reduced to 18°C and 0.2 mM IPTG was added for induction and the culture was grown for ~12 h. The bacterial cells were harvested by centrifugation 5,000 rpm at 5°C and cell pellets were stored at −20°C for further use.

For purification, the cell pellet was resuspended in ice cold lysis buffer containing 50 mM Tris-HCl, 500 mM KCl, 10% (v/v) glycerol at pH 7.5 and a protease inhibitor cocktail (Roche, Switzerland). The cell-suspension was incubated on ice for 1 h with lysozyme and was subsequently lysed by sonication in four pulses of 30 s each and amplitude of 30% interspersed with intervals of 10 s. The crude cell extract obtained was centrifuged at 8,000 rpm for 90 min at 6°C (Sorvall RC-5B Plus Superspeed Centrifuge, Thermo Fisher Scientific, GSA rotor). The supernatant was loaded on a Ni-NTA column pre-equilibrated with 50 mM Tris-HCl, 500 mM KCl; 10% (v/v) glycerol at pH 7.5 and extensively washed with the same buffer containing 0, 10 and 20 mM imidazole. The protein was eluted by imidazole concentrations between 50 and 500 mM. Sample purity was determined by sodium dodecyl sulfate polyacrylamide gel electrophoresis (SDS-PAGE). The nsP2[pro] protein containing fractions were pooled and concentrated using an Amicon Ultra-15 concentrator with a cutoff of ~10 kDa and the partially purified nsP2[pro] sample was loaded onto a Superdex 75 HR 10/30 size exclusion column (GE Healthcare, USA), pre-equilibrated with 20 mM Tris-HCl, 150 mM NaCl, 5% (v/v) glycerol, at pH 7.5.

### Inhibition assay of virus proteases

**ZIKV NS2B/NS3[pro].** Inhibition of Zika virus NS2B/NS3[pro] activity by HST (Sigma Aldrich, USA, purity > 95%) and HSD (Sigma Aldrich, USA, purity > 97%) was studied using the assay described previously [41–43]. Activity of the protein was evaluated using a fluoro-genic substrate (Pyr-Arg-Thr-Lys-Arg-AMC; BACHEM, Bubendorf, Switzerland). The assay was performed in buffer containing 20 mM Tris-HCL, pH 8.5, 10% glycerol, 0.01% Triton X-100. Three nM of protein was incubated with 0–140 μM HST (Sigma-Aldrich, USA) for 1h at RT in the absence of light. The reaction mixture was pipetted into a Corning 96-Well plate (Sigma-Aldrich, USA). When the substrate with a final concentration of 20 μM was added to

the mixture, the fluorescence intensities were measured at 60 s intervals over 10 minutes using an Infinite 200 PRO plate reader (Tecan, Männedorf, Switzerland). The temperature was set to 37˚C, and the excitation and emission wavelengths were 380 nm and 460 nm, respectively.

Activity and inhibition of both investigated proteases was calculated using Eq (1)

% protease activity

$:$ (Intensity of enzyme activity$-$intensity left after inhibition)/Intensity of enzyme activity (1)

All inhibition assays (ZIKV NS2B/NS3$^{pro}$ and CHIKV nsP2$^{pro}$) were performed as triplicates and the results are shown as mean ± standard deviation (SD). Each experiment was performed with fresh purified protein. To determine the half-maximal inhibitory concentration (IC$_{50}$), a dose-response curve was plotted and the IC$_{50}$ determination was based on nonlinear regression [41, 44].

**CHIKV nsP2$^{pro}$.** The protease activity of CHIKV nsP2$^{pro}$ was measured by a FRET-based assay [45] using the commercially synthesized fluorogenic peptide DABCYL-Arg-Ala-Gly-Gly-↓Tyr-Ile-Phe-Ser-EDANS (BACHEM, Bubendorf, Switzerland). This substrate peptide consists of eight amino acids derived from the cleavage site present between non-structural proteins nsP3-nsP4 of CHIKV. The enzymatic reactions of CHIKV nsP2$^{pro}$ were performed in a Corning 96-Well plate having 100 μl reaction volume in 20 mM Bis-Tris-Propane, pH 7.5 as assay buffer. For the investigation of CHIKV nsP2$^{pro}$ inhibition, HST and HSD (Sigma-Aldrich, USA) were tested. 0.5 mM stock solutions of the ligand were prepared by dissolving them in sterile 100% DMSO (Sigma-Aldrich, USA). The final concentrations of inhibitors required were obtained by dilution in the assay reaction buffer. The concentrations of DMSO in the final reaction did not exceed 1.0% (v/v). CHIKV nsP2$^{pro}$ inhibition assay was performed *in vitro* using purified CHIKV nsP2$^{pro}$ pre-incubated with different inhibitor concentrations. Thus, one μM of the protein was incubated with 0–30 μM HST or 0–45 μM HSD for 1h at RT in absence of light. The reaction mixture was pipetted in a Corning 96-Well plate (Sigma Aldrich). When the substrate with a final concentration of 3 μM was added to the mixture, the fluorescence intensities were measured at 60 s intervals over 60 minutes using an Infinite 200 PRO plate reader (Tecan, Männedorf, Switzerland). The temperature was set to 25˚C. The excitation and emission wavelengths were 340 nm and 490 nm, respectively.

## Determination of inhibition mode

The inhibition assay was used to determine the inhibition mode of the tested ligands for each virus protease. Three nM ZIKV NS2B/Ns3$^{pro}$ was pre-incubated with the inhibitor at different concentrations for 60 min at RT (HST: 0–5 μM). Subsequently, the reaction was initiated by addition of the corresponding concentration series of the substrate (0–50 μM in 5 μM steps).

CHIKV nsP2$^{pro}$ at one μM was pre-incubated with the inhibitor at different concentrations for 60 min at RT (HST: 0–5 μM and HSD: 0–20 μM). Subsequently, the reaction was initiated by addition of the corresponding concentration series of the substrate (0, 0.2, 0.4, 0.8, 1.0 and 2.0 μM). All measurements were performed in triplicate and data are presented as mean ± SD. The data analysis was performed using a Lineweaver-Burk plot, therefore the reciprocal of velocity (1/V) vs the reciprocal of the substrate concentration (1/[S]) was compared [46, 47].

## Determination of ligand efficiency

To determine the ligand efficiency for HST and HSD the formula (2) was used [48]:

$$(1.4*p\text{IC}_{50})/\text{N} \tag{2}$$

while $pIC_{50}$ was obtained from an online tool [49] and N is the number of all atoms exception hydrogen.

## Fluorescence spectroscopy

The intrinsic Trp fluorescence was measured with a QuantaMaster40 spectrofluorometer (PTI, Birmingham, USA) using 3 mm path length quartz cuvettes (105.253-QS, Hellma, Mühl-heim, Germany). All spectra were corrected for background intensities by subtracting the spectra of pure solvent measured under identical conditions. Both excitation and emission bandwidths were set at 8.0 nm. The excitation wavelength at 295 nm was chosen since it provides no excitation of tyrosine residues. The emission spectrum was collected in the range of 300–500 nm with the increment of 1 nm. Each point on the emission spectrum is the average of 10 accumulations. The tested virus protease solutions (ZIKV NS2B/NS3$^{pro}$, and CHIKV nsP2$^{pro}$) had concentrations of 5 μM and the measuring volume was 50 μl. The buffer of the experiment contained 25 mM Tris-HCL, pH 8.5, 150 mM NaCl and 5% glycerol.

During the investigation of the ligands interaction with the target, the protein solution within the cuvette was titrated stepwise with a ligand stock solution (0.5 mM ligand + 10 μM protein), ZIKV NS2B/NS3$^{pro}$-HST (0–150 μM), ZIKV NS2B/NS3$^{pro}$-HSD (0–150 μM), CHIKV nsP2$^{pro}$-HST (0–206 μM) and CHIKV nsP2$^{pro}$-HSD (0–206 μM) and after each titration, a measurement was conducted. The quenching of the protease fluorescence, $\Delta F$ ($F^{max}$-F), at 330 nm of each titration point was used for fitting a saturation binding curve using a nonlinear least-squares fit procedure which has been discussed in detail elsewhere [50], based on Eq (3) [51]:

$$Y = B_{max}[Q]/K_D + [Q] \tag{3}$$

where, [Q] is the ligand concentration in solution, acting as a quencher, y is the specific binding derived by measuring fluorescence intensity, $B_{max}$ is the maximum amount of the complex protease-ligand at saturation of the ligand and $K_D$ is the equilibrium dissociation constant. The fluorescence intensity maximum is plotted against the ligand concentration.

To determine the $K_D$ value, the data were fitted with a modified Hill equation obtaining the following relation (4) [52, 53]:

$$Log\ (F-F^{min})/(F) = m\ log\ K_D + n\ log\ [Q] \tag{4}$$

where, $F^{min}$ is the minimal fluorescence intensity in the presence of the ligand, $K_D$ is the equilibrium constant for the protein-ligand complex. The "binding constant" K is defined as the reciprocal of $K_D$, m is the Hill's coefficient and n is the number of occupied binding sites. The fitted data is plotted against the Log of the ligand concentration and the intersection with the x-axis correspond to the $K_D$ value. The two Eqs (3) and (4) were used independently to determine the $K_D$ values.

The measurement of thermal unfolding of the proteases with and without the ligands was performed in the range of 20–85°C with increments of 5°C. Before the experiments started 10 μM of the protease was incubated with 75 μM of the corresponding ligand for 30 minutes at room temperature. The native protein fraction (fN) was calculated according to the following Eq (5):

$$fN = (F-FU)/(FN-FU) \tag{5}$$

where, F, FN and FU are the steady-state fluorescence intensities of Trp at each temperature investigated, at the first (native protein) and the last (unfolded protein) temperature, respectively. The unfolded protein fraction can be calculated as follows: fU = 1 − fN. The temperature

at which fN = fU is called melting temperature or transition midpoint ($T_m$). A one-step denaturation process was assumed.

## Cell lines and viruses

Vero cells (Cercopithecus aethiops kidney normal; ATCC CCL-81) were maintained in Eagle's Minimum Essential Medium (MEM; Sigma-Aldrich) supplemented with 100 U/mL of penicillin (Hyclone Laboratories, United States), 100 mg/mL of streptomycin (Hyclone Laboratories, United States), and 1% of fetal bovine serum (FBS; Hyclone Laboratories, United States) in a humidified 5% $CO_2$ incubator at 37˚C.

A low passage of ZIKVBR58 [54] and CHIKV from a viremic Brazilian patient were used for the antiviral assay. To determine viral titer, $5 \times 10^5$ Vero cells were seeded in each well of a 24 well plate 24 h prior to the infection. Then, the cells were infected with 10-fold serially dilutions of CHIKV or ZIKV for 1 h at 37˚C. The virus inoculum was removed, following the addition of an overlay media containing 1.5% w/v carboxymethylcellulose (Synth, São Paulo, Brazil) in 1% FBS v/v MEM. Infected cells were incubated for three days in a humidified 5% $CO_2$ incubator at 37˚C, followed by fixation with 4% formaldehyde and stained with 0.5% violet crystal. The viral plaques were counted to determine virus titer. Results were expressed as PFU/mL.

## Cell viability through MTT assay

Cell viability was measured by MTT [3-(4,5-dimethylthiazol-2-yl)-2,5-diphenyl tetrazolium bromide] (Sigma-Aldrich) assay. Vero cells at the density of $10^4$ were cultured in each well of 96 well plates and treated with different concentrations (2.3 μM– 300 μM) of each molecule at 37˚C with 5% of $CO_2$. DMSO was used as vehicle control. 48 h post treatment, compound-containing media was removed and MTT solution at 1 mg/mL was added to each well, incubated for 1 h and replaced with 100 μL of DMSO (dimethyl sulfoxide) to solubilize the formazan crystals. The absorbance was measured at 560 nm on Glomax microplate reader (Promega). Cell viability was calculated according to Eq (6):

$$(T/C) \times 100\% \tag{6}$$

in which T and C represented the optical density of the treated well and control groups, respectively. The MTT assays for HST and HSD were performed as triplicates and the results are shown as mean ± SD.

## Antiviral assay

The antiviral assay was performed in 24-well plates with 80,000 Vero E6 cells/well, using a fixed virus inoculum (~50 PFU), following overlay with MEM +1% carboxymethylcellulose (CMC) with or without molecules. Cells were incubated for three days at 37˚C in a humidified atmosphere with 5% $CO_2$. After incubation, overlay medium was removed and the cells were fixed and stained with Crystal Violet. Lysis plaques were counted and compared with non-infected and virus-infected controls. The antiviral assays for HST and HSD were performed as triplicates and the results are shown as mean ± SD.

## Statistical analysis

All experiments underwent at least three independent repetitions and all data are expressed as the means ± the standard deviations (SDs). The statistical significances of the differences in the mean values were assessed with one-way analyses of variance (ANOVA), followed by

Tukey's multiple comparison test. Significant differences were considered at $p < 0.05$ (*), $p < 0.01$ (**) and $p < 0.001$ (***). All statistical analyses were performed with GraphPad Prism software version 5 (San Diego, CA, USA).

## Docking and molecular dynamics simulation

Docking calculation was performed for the proteins-ligands complex using AutoDock Vina 1.1.12 [55]. The AutoDockTools program [56] was used to add polar hydrogens and partial charges to the proteins and to set the rotational bonds of the ligands. Search space was defined using a grid near the protease active site. Several poses for the protein-ligand were ranked according to the scoring function of Autodock Vina. The selected protein-ligand complex was used to start MD simulations.

MD simulations were performed using Amber18 [57] software package. To describe all-atom protein interactions, the FF14SB [58] force field was applied, while the ligands were described using the GAFF and RESP charges. The structures of the proteins were obtained from the PDB database: ZIKV (PDB entry: 5LC0) and CHIKV (PDB entry: 3TRK). The initial protonation state for the amino acids side chain of the proteins were settled using H++ web server [59] at pH 7.5. The systems were neutralized using $Cl^-$ or $Na^+$ ions and placed in an octahedral box with TIP3P water extended 10 Å away from the protein. Bad contacts in the initial structure were removed in two steps of energy minimization. The complex (protein-ligand) was constrained and the force constant of 50.0 kcal/mol-Å$^2$ were performed for 5000 steps of steepest descent followed by 5000 conjugate gradient steps. Subsequently, 10000 steps of an unconstrained energy minimization, afterward, the systems were heated from 0 to 298 k under constant atom number, volume and temperature (NVT) ensemble, while the protein-ligand complex was restrained with constant force of 10 kcal/mol.Å$^2$. Subsequently, the equilibration step was performed using the constant atom number, pressure and temperature (NPT) ensemble for 500 ps and the simulation was performed for 200 ns with 2 fs time step. The constant temperature (298 K) and pressure (1 atm) were controlled by Langevin coupling. Particle-Mesh Ewald method (PME) was utilized to compute the long-range electrostatic interactions and the cut-off distance of 10 Å was attributed to Van der Waals interactions [60].

## Molecular dynamics analyses

The results of the MD simulations were analyzed using CPPTRAJ program [61] of the AmberTools18 package. The root mean square deviations (RMSD) were used to determine equilibration of the systems and convergence of the simulations. Protein flexibility was studied by root mean square fluctuation (RMSF) of the Cα atoms, where RMSF were calculated residue-by-residue over the equilibrated trajectories. Radius of Gyration (RoG) and surface area were calculated to assess major structural changes in the protein. The interaction energy was calculated using the generalized Born (GB)-Neck2 [62] implicit solvent model (igb = 8). Molecular mechanics/generalized Born surface area (MM/GBSA) energy was calculated between the protein and the ligand in the stable regime comprising the last 10 ns of the MD simulation stripping all the solvent and ions.

## Results and discussion

NS2B/NS3 (ZIKV) and nsP2 (CHIKV) proteases are well characterized; both biochemically and structurally and their several specific inhibitors have been previously reported. In this study, we investigated the anti-ZIKV activity of two related compounds derived from citric fruit. These compounds are called Hesperedin (HSD) and its aglycone form Hesperitin (HST), which have been described as antioxidant and anti-inflammatory agents. Both compounds

exhibited antiviral activity against CHIKV protease while the aglycone form was able to inhibit the protease activity of ZIKV.

## Preparation of ZIKV NS2B/NS3$^{pro}$ and CHIKV nsP2$^{pro}$

ZIKV NS2B/NS3$^{pro}$ and CHIKV nsP2$^{pro}$ were expressed in *E. coli* T1 (DE3) cells and purified using Ni-NTA. The relevant protein fractions were concentrated and applied onto a Superdex 200 10/300 GL size exclusion chromatography column (GE Healthcare) in order to remove *E. coli* contamination and aggregated protein species and the purity of the proteases was checked by SDS-PAGE 15% (S1 and S2 Figs, respectively).

## Inhibitory potential of HST and HSD against zika and chikungunya virus proteases

The increased presence of flavivirus and alphavirus in co-infections and due to lack of appropriate medications to combat such infections makes it necessary to identify inhibitors or lead compounds that can control and prevent these co-infections.

In this context, the inhibitory effect of HST and HSD (20 µM) was tested against NS2B/NS3$^{pro}$ of ZIKV and against nsP2$^{pro}$ of CHIKV. A reduction in the enzymatic activity of the tested proteases by HST was observed. HSD showed a strong inhibitory effect on CHIKV nsP2$^{pro}$ (Fig 2) while a low effect against ZIKV NS2B/NS3$^{pro}$, hat agrees with the results described previously [40].

Based on these results, further experiments using different concentrations of HST and HSD were performed. HST has been tested against ZIKV NS2B/NS3$^{pro}$ (0–140 µM) (Fig 3A) and both the molecules were tested against nsP2$^{pro}$ of CHIKV at concentration ranges of 0–30 µM (HST) and 0–45 µM (HSD) (Fig 4A and 4C). The titration of the inhibitors demonstrated the inhibitory strength of the compounds and were comparable, even at low inhibitor

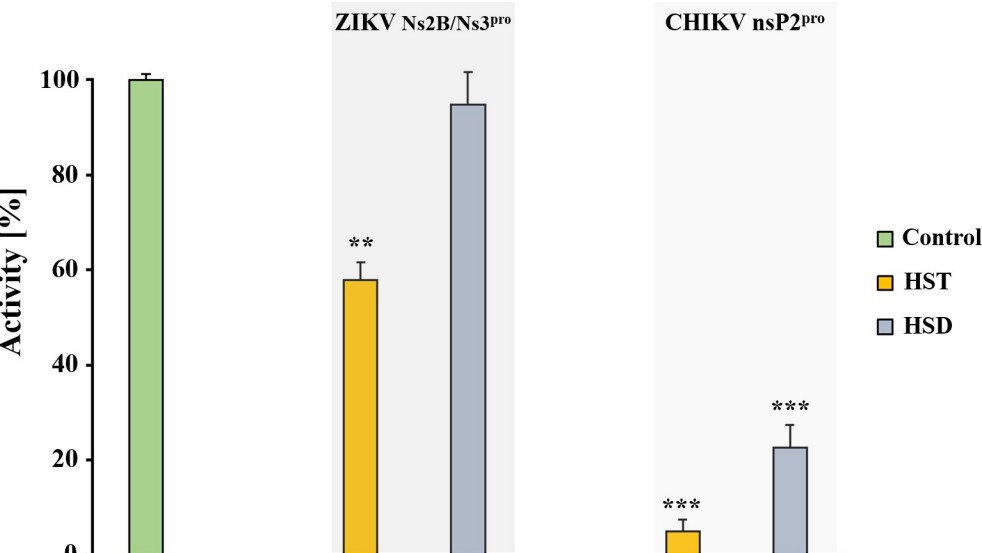

**Fig 2. Preliminary inhibition test of HST and HSD against ZIKV NS2BNS3$^{pro}$ and CHIKV nsP2$^{pro}$.** The tested compound concentration was 20 µM. HST inhibit ZIKV NS2B/NS3$^{pro}$ activity about 40% and CHIKV nsP2$^{pro}$ about 90%. The inhibition of CHIKV protease activity by HSD was higher than 60%. Contrary, HSD shows not relevant inhibition against ZIKV NS2B/NS3$^{pro}$. Data shown are the means ± SD from three independent measurements (n = 3). Asterisks mean that the data differs from the control (0 µM inhibitor) significantly at $p < 0.01$ (**) and $p < 0.001$ (***), level according to ANOVA and Tukey's test.

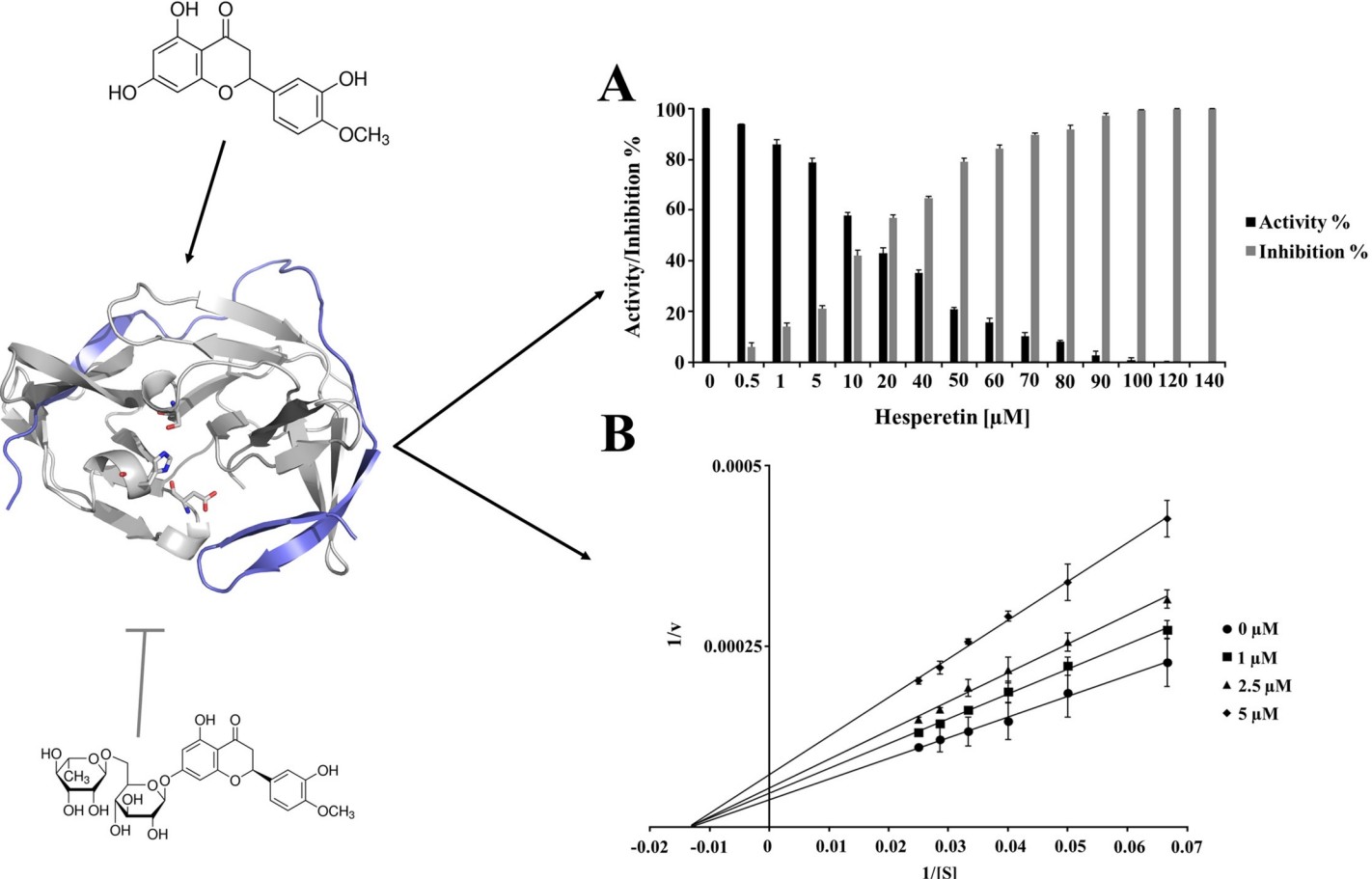

**Fig 3. Inhibition effect and inhibition mode of HST on ZIKV NS2B/NS3^pro.** Chemical structure of HST and HSD (structural formula) and ZIKV NS2B/NS3^pro 3D experimental model (PDB entry: 5LC0) are shown. NS3^pro is represented in gray and NS2B in blue. Normalized activity and inhibition of the virus proteases and Lineweaver-Burk plots to determine the inhibition mode is presented: [S] is the substrate concentration; v is the initial reaction rate. The data shown are the means ± SD from three independent measurements (n = 3). **A:** Normalized activity and inhibition of ZIKV NS2B/NS3^pro under HST influence. **B:** Lineweaver-Burk plot for HST inhibition of ZIKV NS2B/NS3^pro.

concentrations. The HST and HSD titration data was used to determine the half-maximal inhibitory concentration ($IC_{50}$) (S3 Fig) of each protease. The results are summarized in Table 1.

To determine the inhibition mode of HST on ZIKV NS2B/NS3^pro, kinetic experiments were performed (Fig 3B). The maximum rate ($V_{max}$) values for the ZIKV protease activity decreased in the presence of different concentrations of HST. On the other hand, the Michaelis constant ($K_m$) values are unaffected. These relationships between $V_{max}$ and $K_m$ indicate a noncompetitive inhibition according to classical Michaelis-Menten kinetics.

HST and HSD showed a noncompetitive inhibition against CHIKV nsP2^pro (Fig 4B and 4D) demonstrating the same inhibition mode as for ZIKV NS2B/NS3^pro.

The $IC_{50}$ value of HST demonstrated a potential inhibitory effect against ZIKV and CHIKV proteases. Additionally, HSD showed a strong inhibitory activity against CHIKV nsP2^pro but not against ZIKV NS2B/NS3^pro.

Roy and collaborators (2017) identified five flavonoids (Myricetin, Quercetin, Isorhamnetin, Luteolin, Apigenin and Curcumin) that inhibit ZIKV NS2B/NS3^pro activity in a noncompetitive mode [46], and we have demonstrated for HST. The calculated $IC_{50}$ values for the five

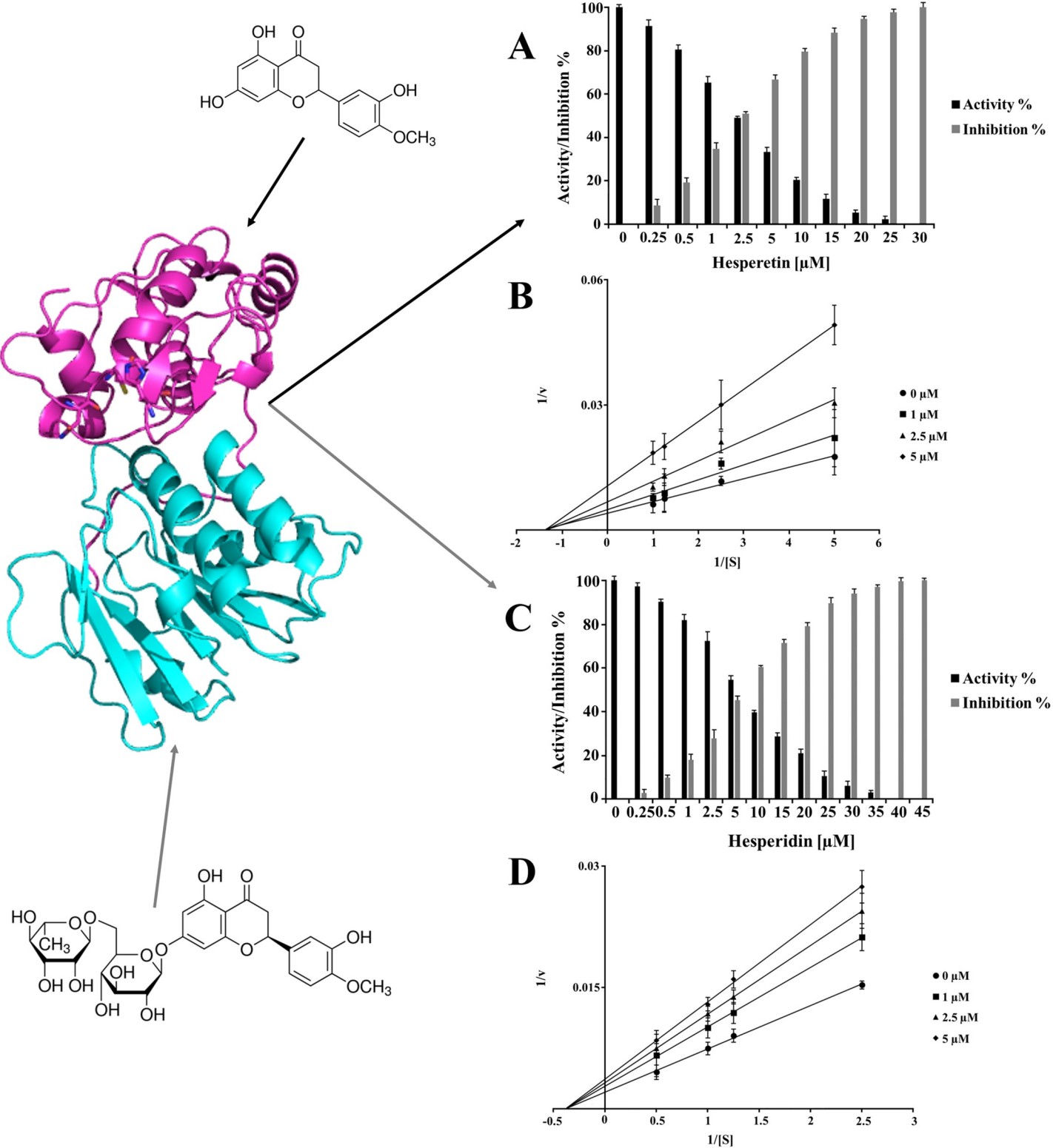

**Fig 4. Inhibition effect and inhibition mode of HST and HSD on CHIKV nsP2^pro.** Chemical structures of HST and HSD (structural formula) and CHIKV nsP2^pro 3D experimental model (PDB code: 3TRK) are shown. CHIKV nsP2 protease domain is presented in pink and the methyltransferase domain in cyan. Normalized activity and inhibition of the virus protease and Lineweaver-Burk plots to determine the inhibition mode is presented: [S] is the substrate concentration; **v** is the initial reaction rate. The data shown are the means ± SD from three independent measurements (n = 3). **A and C:** Normalized activity and inhibition of CHIKV nsP2^pro under HST and HSD influence, respectively. **B and D:** Lineweaver-Burk plot for HST and HSD inhibition of CHIKV nsP2^pro, respectively.

**Table 1. Results summarize HST and HSD inhibition experiments of ZIKV NS2B/NS3$^{pro}$, and CHIKV nsP2$^{pro}$.**

| | Molecule | IC$_{50}$[a] | Inhibition mode | pIC$_{50}$[b] | LE[c] | N[d] | LogP[e] |
|---|---|---|---|---|---|---|---|
| **ZIKV NS2B/NS3$^{pro}$** | HST | 12.6 ± 1.3 | noncompetitive | 4.9 | 0.31 | 22 | 2.6 |
| * [46] | Myricetin | 1.3 ± 0.1 | noncompetitive | | | | |
| | Quercetin | 2.4 ± 0.2 | noncompetitive | | | | |
| | Luteolin | 2.7 ± 0.3 | noncompetitive | | | | |
| | Isorhamnetin | 15.5 ± 0.7 | noncompetitive | | | | |
| | Apigenin | 56.3 ± 0.9 | noncompetitive | | | | |
| | Curcumin | 3.5 ± 0.2 | noncompetitive | | | | |
| **CHIKV nsP2$^{pro}$** | HST | 2.5 ± 0.4 | noncompetitive | 5.6 | 0.36 | 22 | 2.6 |
| | HSD | 7.1 ± 1.1 | noncompetitive | 5.2 | 0.17 | 43 | -0.3 |
| ** [63] | Pep-I | 34 | noncompetitive | | | | |
| | Pep-II | 42 | - | | | | |
| *** [64] | Novobiocin | 2 | - | | | | |
| | Telmisartan | 5 | - | | | | |

[a]IC$_{50}$ value in μM.

[b]Logarithm of IC$_{50}$ value (pIC50).

[c]Ligand efficiency (LE).

[d]Number of non-hydrogen atoms (N).

[e]Information about hydrophobicity of the molecule (A negative value for logP means that the compound has a higher affinity for the aqueous phase (it is more hydrophilic). When LogP = 0 the compound is equally partitioned between the lipid and aqueous phases; a positive value for LogP denotes a higher concentration in the lipid phase (i.e., the compound is more lipophilic).

flavonoids are shown in Table 1 and we have demonstrated that the IC$_{50}$ value for HST was in the same range as that of other flavonoids.

Singh and collaborators (2018) showed IC$_{50}$ values of 40 μM for two peptides studied against CHIKV nsP2 protease, designated as Pep-I and Pep-II [63]. However, HST and HSD were consistently more effective in inhibiting the same protease with IC$_{50}$ values of 2.5 ± 0.4 (HST) and 7.1 ± 1.1 (HSD), thereby demonstrating the efficiency like other flavonoid molecules to inhibit the enzyme activity. On the other hand, our IC$_{50}$ values are in a comparable range to Novobiocin and Telmisartan with 2 and 5 μM, respectively (Table 1) [64].

To get more information on the promising inhibitors and to compare them to the already determined inhibitors, we calculated ligand efficiency values of HST and HSD. The ligand efficiency (LE) was determined to validate the ratio of the ligand binding affinity versus the number of non-hydrogen atoms (N) of both compounds (Table 1). For HST, the smaller molecule, the efficiency value was > 0.30. The identified LE values of HST showed it to be an efficient binder [48], demonstrating its potential to be used as a lead compound for further development as a broad-spectrum drug to combat ZIKV and CHIKV infections. In contrast, LE for HSD are under the threshold for the method, dependency on ligand size have been observed in the literature [65]. Chemical modification of HST and HSD could improve the inhibitor efficiency.

## Flavivirus and alphavirus protease interaction studies with HST and HSD using fluorescence spectroscopy

For the investigation of the molecular interaction with the proteases, intrinsic tryptophan (Trp) fluorescence was measured. Based on the quenching of the proteases fluorescence at each titration point, dissociation constant (K$_D$) could be determined for HST- and HSD- with the target proteins (S4 and S5 Figs). The calculated K$_D$ values are summarized in Table 2.

**Table 2. Results summary of the fluorescence spectroscopy experiments of ZIKV NS2B/NS3$^{pro}$ and CHIKV nsP2$^{pro}$ with HST and HSD.**

| | Molecule | $K_D$[a] | $\Delta Tm$[b] |
|---|---|---|---|
| **ZIKV NS2B/NS3$^{pro}$** | HST | 17.8 ± 2.9 | 6 |
| **CHIKV nsP2$^{pro}$** | HST | 31.6 ± 2.5 | 8 |
| | HSD | 40.7 ± 2.0 | 5 |

[a]$K_D$ value ± STD in μM.

[b]$\Delta Tm$ in ˚C.

HST showed an acceptable affinity, for lead molecule, for the interaction with the ZIKV protease (17.8 ± 2.9 μM). In contrast, the $K_D$ value of CHIKV nsP2$^{pro}$-HST increased approx. two folds, 31.6 ± 2.5 μM, which revealed a lower affinity and the lowest interaction, was observed for CHIKV nsP2$^{pro}$-HSD with 40.7 ± 2.0 μM.

The thermal denaturation of ZIKV NS2B/NS3$^{pro}$, ZIKV NS2B/NS3$^{pro}$-HST, CHIKV nsP2$^{pro}$, CHIKV nsP2$^{pro}$-HST, and CHIKV nsP2$^{pro}$-HSD was studied by fluorescence spectroscopy (S6 and S7 Figs). The highest change in the melting temperature ($\Delta T_m$) was observed for CHIKV nsP2$^{pro}$-HST complex (about + 8˚C, Table 2) when compared with the native protein. The increase in thermal stability after ligand binding can be based on structural changes induced by the ligand interaction. A similar trend has been observed in the literature, where an increase in thermal stability (about + 3.6˚C) after the binding of a dipeptide inhibitor with ZIKV NS2B/NS3$^{pro}$ was observed [66].

## Docking and molecular dynamic simulations studies of HST and HSD with flavivirus and alphavirus protease 3D structures

To investigate the dynamic and structural changes of the target proteins in the presence of inhibitory molecules, experimental models of ZIKV NS2B/NS3$^{pro}$ (PDB entry: 5LC0) and CHIKV nsP2$^{pro}$ (PDB entry: 3TRK) structures were submitted, separately, to 200 ns of molecular dynamics (MD) simulations (just single molecules). Two independent simulations for each structure were performed. The structural mobility of the system was monitored by calculating the RMSD, RMSF, RoG and the surface area. The standard Cα RMSDs of the two simulations for ZIKV and CHIKV proteases are represented in S8 and S9 Figs, respectively. In simulation I (S8A Fig), the Cα RMSD is below 2Å for most of the simulation. The NS2BNS3 protease structure of ZIKV, therefore, is very stable, and does not diverge from the initial structure. In simulation II the standard RMSD (S8B Fig) shows a greater fluctuation with RMSD oscillating between 1.5 and 2.5, however, it becomes lesser than 2.5 Å at about 50 ns. Clearly, the structure undergoes some structural change. In case of nsPs protease structure of CHIKV, both simulations demonstrated relatively similar pattern with RMSD below 2.4 Å (S9 Fig).

The S1-S3 pockets of the ZIKV protease are very well defined; that show a preference for a substrate with dibasic residues (P1-P2). Besides, a hydrophobic allosteric pocket has been mapped for ZIKV protease, located near the NS2B cofactor interaction area. The activity test results demonstrated that HST and HSD are noncompetitive inhibitors for both proteases (ZIKV NS2B/NS3 and CHIKV nsP2), which means, that the ligand inhibit the protein allosterically. To identify a possible allosteric binding site we performed molecular docking and MD simulations. The noncompetitive inhibitors were docked near the protease active site, and two sets of simulations were performed and analysis of the protein structural fluctuation of both simulations in complex with the ligand molecules improved the protein stability over the 200 ns (S10–S12 Figs).

The susceptibility of secondary structural content is an essential component to study the structural behavior of the protein under external influence. Analysis of the ZIKV and CHIKV protease complexes with the tested molecules indicated minor variations in the secondary structure contents and length (S13 and S14 Figs). However, we suggest that the binding of HST to the allosteric site of ZIKV NS2B/NS3$^{pro}$ slightly changes the secondary structure elements inducing conformational changes (S13 Fig). As the secondary structure is a trigger for the protease activity, those changes might influence in the spatial conformation of some amino acids residues, as example, the Asp75 (S13 Fig), one of the catalytically residue, which may inactivate the protease activity.

The binding of HST and HSD to the nsp2$^{pro}$ (CHIKV) induce a small structural elements "adjustment" near to the active residue His81. We assume that the alteration of the His81 position may disturb the active site construction and consequently the function (S14 Fig). These variations in the protein secondary structure can play a critical role in the thermal stability [67, 68] that can explain the $\Delta T_m$ results described before.

Amino acid residues, which are responsible for protein-ligand interaction were identified using the interaction energy. Fig 5 illustrates the decomposition of the binding energy as well as the amino acids residues of the ZIKV protease that form the complex with HST. The Asn152 residue forms a hydrogen bond with the inhibitor. The other residues showed in the Fig 5B are involved in hydrophobic interaction with the inhibitor molecule. Fig 6 displays the decomposition of the binding energy (Fig 6A and 6B) and the interaction between nsP2$^{pro}$-HST (Fig 6C) and–HSD (Fig 6D) for the protease of CHIKV. For the analyses, we have used the outcome of the two independent MD simulation runs Amino acids that form hydrogen bonds with the ligands, but appear only once in the analysis, were not considered for further investigation.

Hydrophobic (Val29, Leu31, Phe37, Thr119, Asp121, Ile124 and Asn152) and hydrogen bonding (Asp 152) interactions were found to be responsible for the stable interaction between HST and ZIKV NS2B/NS3$^{pro}$. Both interactions are key players in stabilizing energetically favored ligands. Several amino acids from both domains are involved in the interaction with HST; three residues of the NS2B domain (Val29, Leu31 and Phe37) and four of the NS3 protease domain (Thr119, Asp121, Ile124 and Asn152) (Fig 5A). Asn152 forms a single hydrogen bond (H-bond) with HST and the side chain NH$_2$ group of Asn152 functions as donor and O1 of HST acceptor (Tables 3 and 4, atom numbers of HST and HSD are shown in S15 Fig). For more details, see Table 3.

The catalytic triad (His51, Asp75 and Ser135) do not interact directly with the ligand and are in agreement with the competition assay results that identified HST as a noncompetitive inhibitor of ZIKV NS2B/NS3$^{pro}$. From our results, we can assume that HST is an allosteric inhibitor, binding at a site other than the active site and this interaction altered the catalytic site conformation, thereby, prevented the entry of the substrate. Additional experiments are needed to confirm this assumption.

The results of two independent MD runs with ZIKV NS2B/NS3$^{pro}$ and HST demonstrate the reproducibility of the HST position and interacting amino acids during 200 ns of simulation (Fig 5C).

Eleven amino acids are involved in the interaction between CHIKV nsP2$^{pro}$ and HST and all of them are located in the cysteine protease domain (Pro47, Glu48, Leu51, Leu63, Val75, Tyr76, Tyr77, Trp82, Gly88, Lys89, Phe91) (Table 3 and Fig 6A). Lys89 form a single H-bond with HST, which is mediated by the backbone NH group of the amino acid (acceptor) and the O5 (OH, donor) of HST (Table 4). Pro47, Glu48, Leu51, Leu63, Val75, Tyr76, Tyr77, Trp82, Gly88 and Phe91 stabilize the binding by hydrophobic interactions.

HSD interact with eight amino acids in the cysteine protease domain (Pro47, Glu48, Val75, Tyr76, Tyr77, Trp82, Lys89, Ph91) and three in the methyltransferase domain (Leu203, Met236, Met240) (Fig 6B).

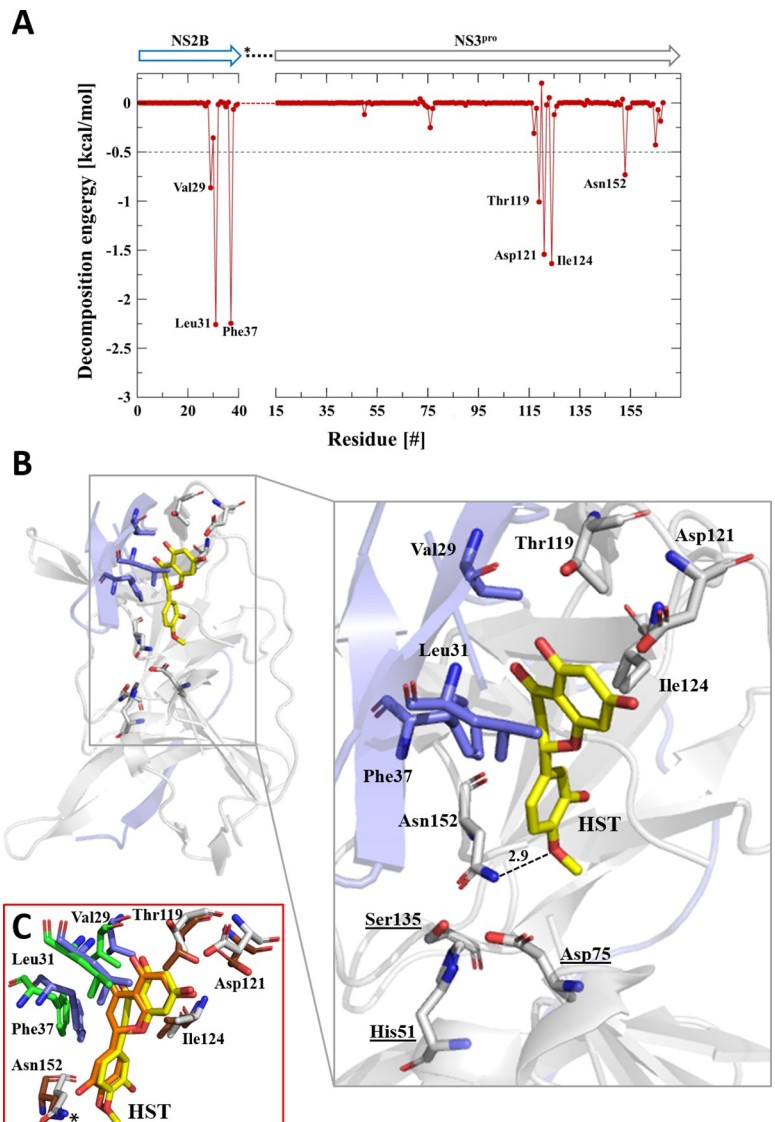

**Fig 5. Amino acids participating in the ZIKV NS2B/NS3^pro^-HST interaction. A:** Decomposition of the binding energy of ZIKV NS2B/NS3^pro^-HST complex. Arrows label NS2B (blue) and NS3^pro^ (gray). The residues involved in the interaction between protease and HST are labeled. **B:** 3D structure of ZIKV NS2B/NS3^pro^ (PDB entry: 5LC0), the NS3^pro^ domain is colored in gray and NS2B in blue. Amino acids highlighted are involved in the interaction with HST (yellow) based on MD simulations. The H-bond between Asn152 and HST is highlighted. **C:** Structural overlay of the representative complex structures of two independent MD runs (RMSD: 1.475 Å). Run1: NS2B (blue) and NS3^pro^ (gray) and run2: NS2B (green) and NS3^pro^ (brown). The binding region of HST is highlighted, the involved amino acids and HST position are the same between run1 and run2. The distance between Asn152 and HST differs slightly between MD run1 and run2 (2.9 to 2.8 Å).

Glu48, Val75 and Tyr77 form H-bonds with HSD and stabilize the interaction (Table 3). The greater number of H-bonds can explain the higher ligand-binding energy observed for the CHIKV nsP2^pro^-HSD complex with -37.1 ± 3.7 kcal/mol, compared with CHIKV nsP2^pro^-HST with -23.5 ± 2.3 kcal/mol and ZIKV NS2B/NS3^pro^-HST with -25.5 ± 4.1 kcal/mol.

The Glu48 backbone (NH group) interacts with O3 of HSD. Val75 and Tyr77 interacts with the OH groups in the sugar moiety of HSD. Val75 backbone O interact with O8 (OH) of HSD and Tyr77 backbone O with O10 (OH) of HSD (Table 4). As described before for ZIKV NS2B/

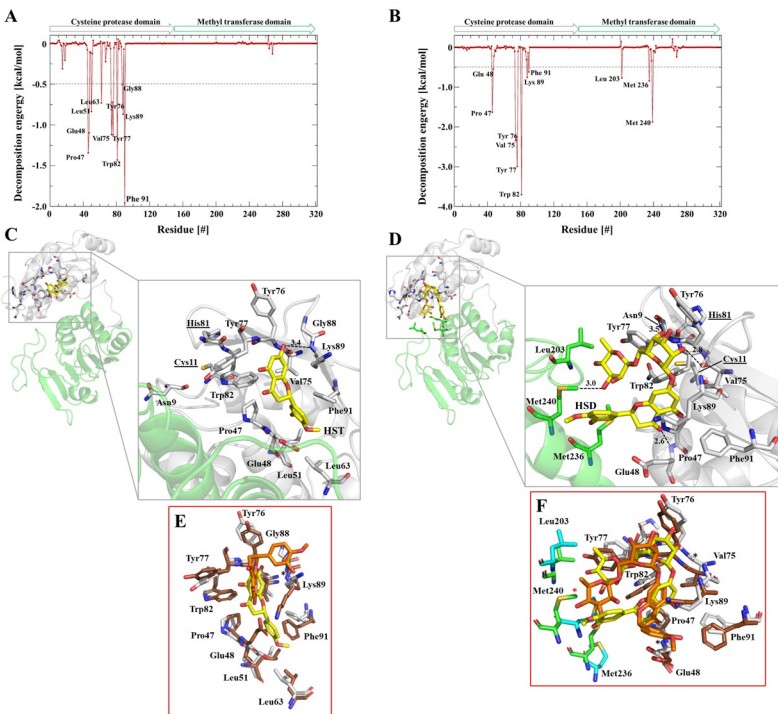

**Fig 6. Amino acids participating in the CHIKV nsP2^pro^-HST and -HSD interaction. A:** Decomposition of the binding energy of CHIKV nsP2^pro^-HST complex. Arrows label the protease domain (gray) and methyltransferase domain (green). The residues involved in the interaction between protease and HST are labeled. **B:** Decomposition of the binding energy of CHIKV nsP2^pro^-HSD complex. The residues involved in the interaction between protease and HSD are labeled. **C:** 3D structure of CHIKV nsP2^pro^ (PDB entry: 3TRK), the protease domain is colored in gray and methyltransferase domain in green. Amino acids highlighted that are involved in the interaction with HST (yellow) based on MD simulations. The H-bond between Lys89 and HST is highlighted. **D:** 3D structure of CHIKV nsP2^pro^, amino acids highlighted are involved in the interaction with HSD (yellow) based on MD simulations. The H-bonds between Tyr77, Met240, Val75 and Glu48 with HSD are highlighted. **E:** Structural overlay of the representative CHIKV nsP2^pro^-HST complex structures of two independent MD runs (RMSD: 1.259 Å). Run1: nsP2^pro^ (gray), HST (yellow) and run2: nsP2 (brown), HST (orange). The binding region of HST is highlighted, the involved amino acids and HST position differs between run1 and run2. In run2 Pro47, Leu51, Leu63 and Ph91 are not interacting by hydrophobic interactions with HST. However, Lys89 forms in run2 like in run1 a hydrogen bond to HST, the distances between donor and acceptor differs slightly between MD run1 and run2 (3.4 to 3.5 Å). **F:** Structural overlay of the representative CHIKV nsP2^pro^-HSD complex structures of two independent MD runs (RMSD: 1.733 Å). Run1: nsP2^pro^ (gray), methyltransferase domain (green), HSD (yellow) and run2: nsP2 (brown), methyltransferase domain (cyan), HST (orange). The binding region of HST is highlighted, the involved amino acids and HST position differs slightly between run1 and run2. The main difference concerns Met240, which forms in run1 a hydrogen bond, in run2 this interaction is not more formed, the distance increased from 3.0 to 5.5 Å. Because of this observation, Met240 will be not further considered. The hydrogen bonds formed by Glu48, Val75 and Tyr77 remain unaffected.

NS3^pro^, in CHIKV nsP2^pro^ the active site residues (Cys11 and His81) do not interact directly with HST and HSD as demonstrated by the noncompetitive inhibition mechanism.

Using computational approaches, the localization of probable allosteric sites in ZIKV NS2B/NS3^pro^ and CHIKV nsP2^pro^ were defined previously [46, 69, 70]. They described that

**Table 3. Residues involved in forming H-bonds and hydrophobic contacts between the proteins and the ligands.**

| Protein | Ligand | Interacting Residue | |
|---|---|---|---|
| | | *H-Bond* | *Hydrophobic interaction* |
| **ZIKV NS2BNS3^pro^** | HST | Asn152 | Val29, Leu31, Phe37, Thr119, Asp121, Ile124 |
| **CHIKV nsP2^pro^** | HST | Lys89 | Pro47, Glu48, Leu51, Leu63, Val75, Tyr76, Tyr77, Trp82, Gly88, Phe91 |
| | HSD | Tyr77, Val75, Glu48 | Pro47, Tyr76, Trp82, Lys89, Phe91, Leu203, Met236, Met240 |

**Table 4. Atoms involved in the H-Bond interaction between flavivirus and alphavirus proteases and ligands.**

| Protein | Ligand | Atom (Amino acid) | Atom (Ligand) | H-bond Donor/Acceptor |
|---|---|---|---|---|
| ZIKV NS2B/NS3[pro] | HST | **Asn152** | | |
| | | NH$_2$ (Side chain) | O1 | **N-H-—-O (2.9)[a]** |
| CHIKV nsP2[pro] | HST | **Lys89** | | |
| | | NH (Backbone) | O5 | **N-—-H-O (3.4)** |
| | HSD | **Glu48** | | |
| | | NH (Backbone) | O3 | **N-H-—-O (2.6)** |
| | | **Val75** | | |
| | | O (Backbone) | O8 | **O-—-H-O (2.8)** |
| | | **Tyr77** | | |
| | | O (Backbone) | O10 | **O-—-H-O (3.5)** |

[a]Distance between H-Bond donor and acceptor shown in Å.

the ZIKV NS2B/NS3[pro] allosteric site are formed by Lys73, Gln74, Asp123, Gly124, Asp125, Ala 128, Asn152 and Ala164 amino acid residues [46, 71]. In our studies, we docked the HST molecule, which is a noncompetitive ligand, based on the same allosteric site. The interaction with ZIKV NS3B/NS3[pro] showed the same amino acids as reported by Roy et al. 2017 [46] and mostly interactions are based on hydrophobic interactions, H-bond can be observed with the amino acid Asn152 (Table 3 and S16A Fig). Allosteric site of CHIKV nsP2 have already been proposed [72], based on the previous results, HST and HSD was docked in the supposed allosteric site, which is formed by Ser46, Glu48, Val49, Val75, Tyr76, Tyr77, Trp82, Gly88, Leu203, Pro206, Met236, Lys237, Gln239, Met240 and Asp244 residues [72]. HST showed interaction with several amino acid residues of the proposed allosteric site of nsP2 protease by hydrophobic interaction (Glu48, Val75, Tyr76, Tyr77, Trp82, Gly88 and Leu203) (Table 3 and S16B Fig), HSD interacted with the same amino acids but showed one additional interaction with Met236 (Table 3 and S16C Fig).

To investigate further the plasticity of the virus proteases in complex with HST and HSD and the possible consequences of the ligand binding, the substrate binding sites S1, S1´, S2, S3, S4 and the oxyanion hole after MD simulation of the complexes were analysed. Amino acids comprising the substrate binding sites of ZIKV NS2B/NS3[pro] and CHIKV nsP2[pro] were previously described [71, 72], and we have presented the results in S2 Table and Fig 7. Conformational changes of both proteases after ligand binding were observed, probably due to allosteric modulation (Fig 7).

In the case of ZIKV NS2B/NS3[pro] a single amino acid, Asn152 (S2), form a H-bond with HST. In case of CHIKV nsP2[pro], the Trp82 (S2 and S4) and the Tyr77 (S3) are involved in the interaction with HST and HSD, whereas Met240 (S3) only with HSD (Fig 7). As described before, HST-Asn152 H-bond probably modify the shape of the substrate binding site. Interestingly, the effect of HST on CHIKV nsP2[pro] seems to follow the same principle. The residues of the S2, S3 and S4 sites of CHIKV nsP2[pro], mediate the HSD interactions. These subsites are no more accessible for the substrate because the interaction with HSD cause changes in the shape of the substrate binding site.

## Pharmacokinetic view on the eligibility of HST and HSD as lead molecules targeting viral proteases

Bioavailability is a key factor in ensuring the efficacy of bioactive flavonoids. Studies demonstrated that HSD had limited bioavailability and required deglycosylation to be absorbed in the

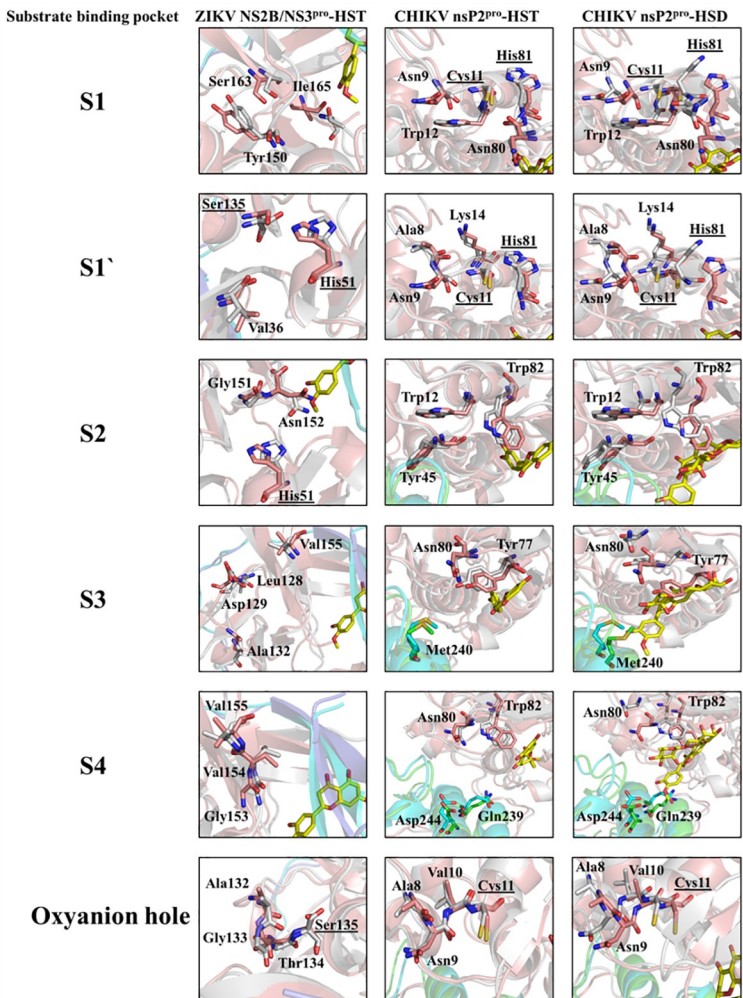

**Fig 7. Substrate binding pockets and oxyanion hole of ZIKV NS2B/NS3$^{pro}$ and CHIKV nsP2$^{pro}$ (S1, S1´, S2, S3, S4) under the influence of HST and HSD.** Structural overlay with the native protein structures, the active site residues are highlighted. The amino acid residues and the inhibitors are shown in sticks.

colon or intestine [73–75]. HSD is converted to HST by the microbiota of the colon, and then, it is absorbed by colonocytes via proton-coupled active transport and transcellular passive diffusion. On the other hand, HST and HST 7-glucoside are directly absorbed by enterocytes [73, 74]. Table 5 summarize the pharmacological kinetics for HST and HSD, described earlier by Nielson et al. and Manach et al. [74, 75].

Another key factor to evaluate the eligibility of potential lead compounds is toxicity or tolerated dose for the cells. Cytotoxicity of HST and HSD was tested by MTT assay on Vero cells (Fig 8)

Vero cells were treated with HST and HSD at concentrations ranging from 2.3 to 300 μM. The results showed that both molecules have no cytotoxic effect at the calculated IC$_{50}$ concentrations, which were determined by *in vitro* experiments. The low cytotoxicity of both molecules is in agreement with results described before [76] (Table 5).

A major concern about protease inhibitors is the negative effect on human proteases. In general, HST and HSD are extremely safe [77]. Shawar et al. 2013 investigated the inhibitory

**Table 5. Pharmacological kinetics for HST and HSD.**

| Pharmacological kinetics* | Hesperetin | Hesperidin |
|---|---|---|
| $T_{1/2}$ (h) | 5.32 ± 0.62 | 6.74 ± 0.79 |
| $C_o$ (μM) | 7.35 ± 1.71 | 7.47 ± 1.64 |
| AUC last (h μM) | 3.01 ± 0.29 | 4.12 ± 0.26 |
| AUC inf. (h μM) | 3.49 ± 0.37 | 4.92 ± 0.38 |
| $V_d$ (L/kg) | 145.60 ± 8.84 | 64.58 ± 3.70 |
| Cl (L/h/kg) | 19.09 ± 1.99 | 6.69 ± 0.58 |
| MRT (h) | 3.60 ± 0.21 | 4.22 ± 0.67 |
| Dose of administration | 20 mg/kg | 20 mg/kg |
| Potential maximal tolerated dose** | Citrus flavonoids are extremely safe and without side effects also up to high concentrations of 100 mg/kg to 500 mg/kg. | |

* [75].

** [76].

$T_{1/2}$ elimination half-life.

$C_o$ initial concentration.

AUC area under the curve.

$V_d$ volume of distribution.

Cl systemic clearance.

MRT mean residence time.

effect of both flavonoids against trypsin and the calculated $IC_{50}$ values were 104 μM (HST) and 127 μM (HSD) [78], respectively. The $IC_{50}$ values for HST and HSD described in our study are 10 to 50 times lower and this concentration will probably not affect trypsin activity. The same applies to the human pancreatic elastase and HST [79].

HST was reported to affect CHIKV intracellular replication using a CHIKV replicon cell-line. In the same study the authors demonstrated that HST possess no antiviral activity against anti-entry, anti-adsorption, direct virucidal assays and the stages of CHIKV replication cycle [29]. Similar results were observed for HSD against CHIKV and HST against ZIKV in plaque reduction assay. Vero cells infected with 50 PFU of CHIKV or ZIKV, after 48 h incubation,

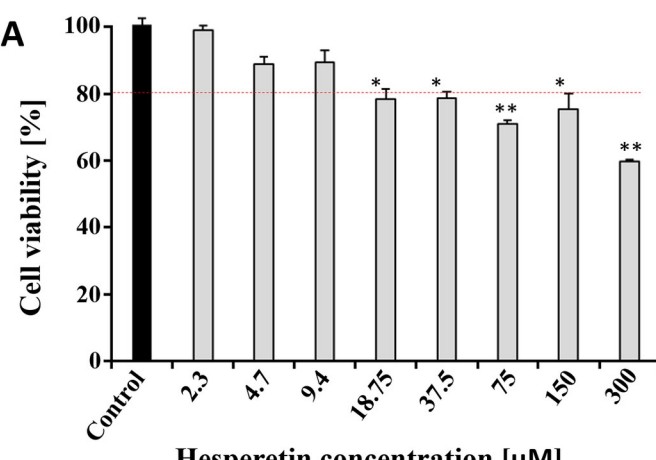
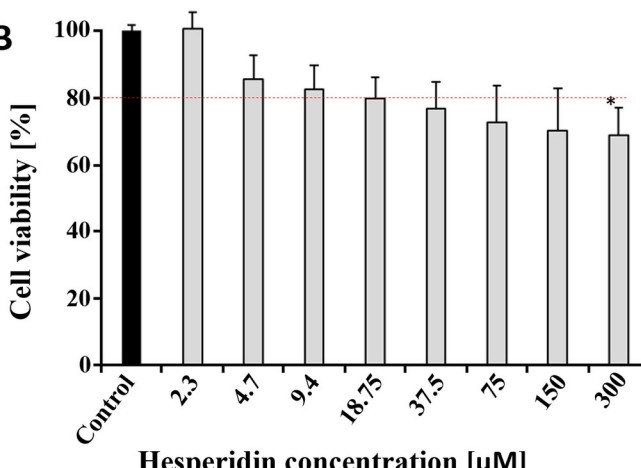

**Fig 8. MTT assay of HST and HSD on Vero cells.** MTT assay was used to evaluate the cytotoxicity of the two compounds. Different concentrations up to 300 μM were used to treat the Vero cells for 2 days. Data shown are the means ± SD from three independent measurements (n = 3). Asterisk means the data differed from the control (0 μM) significantly at p < 0.05 (*) and at p < 0.01 (**), level according to ANOVA and Tukey's test. **A:** Hesperetin and **B:** Hesperidin.

were challenged with two different HST or HSD concentrations. No inhibition was detected in these samples (S17 Fig).

## Conclusion

Globally, arbovirus infections are increasing in numbers, which cause common severe febrile diseases, that can lead to long-term physical or cognitive impairment with severe consequences, where in some cases cause early deaths have been reported [2, 7]. More than 100 arboviruses are identified to cause disease in humans and recently, the co-infections caused by ZIKV and CHIKV pose a new challenge to combat the diseases.

Our results report the use of a natural molecule that can be used as a lead compound against both viruses proteases (ZIKV and CHIKV), to combat their co-infections. Flavivirus and alphavirus proteases are members of different families and the discovery of natural compounds affecting serine and cysteine proteases with a high efficiency is an achievement. Our results showed that HST has a strong inhibitory activity on viral proteases with the $IC_{50}$ values $< 13$ μM. The resulting ligand efficiencies ($> 0.3$) indicate typical values of lead compounds [44] that allow us to infer the importance of this molecule for future studies. The activity assays identified HST as a noncompetitive inhibitor and docking and MD studies exhibited the hypothetical binding site. Even though the molecule is not "touching" the active site triad of the proteases; we assume that the interaction in the allosteric site could modify the substrate-binding site. Furthermore, even though exhibiting a weak inhibitory effect against ZIKV protease, HSD has the potential to inhibit CHIKV nsP2$^{pro}$ strongly with an $IC_{50}$ value lower than 10 μM. The sugar moiety of HSD seems to be the major portion of the molecule that may be restricting the interaction with ZIKV protease, but positively affecting the interaction with CHIKV nsP2$^{pro}$. The partly deglycosylation of HSD in the human gut to HST generate a mixture of both molecules and from our results we can assume that HST and a mixture of HST and HSD could inhibit both proteases strongly.

MTT assays demonstrated that HST and HSD have no cytotoxic effect at the calculated $IC_{50}$ concentrations, which were determined by *in vitro* experiments. However, plaque reduction assays showed no antiviral effect of HST and HSD against both viruses.

However, identification of a lead compound is the first step in making a new drug to treat a disease. Our results confirm that HST and HSD inhibit virus proteases at low μM concentrations and the binding affinities were $< 45$ μM. Additionally, both molecules possess a very low cytotoxicity. Once a lead compound has been found, the chemical structure is used as a starting point to improve the properties of it (by chemical modifications) and design a drug that is most potent and the least harmful to host. Essential modifications can enhance the relevant pharmaceutical properties of the drug molecules such as inhibition and affinity with the targets; specificity against the targets; uptake of the molecules in the cell and cell wall passage; metabolism of the molecules in the cell.

## Supporting information

**S1 Fig. Preparation of ZIKV NS2B/NS3$^{pro}$.** Expression and purification of ZIKV NS2B/NS3$^{pro}$. The ZIKV NS2B/NS3$^{pro}$ construct consists of 266 amino acids with a molecular weight of 28.68 kDa. The protein presented a single band on a denaturing SDS-PAGE gel with an apparent molecular mass of approximately 30 kDa. **A:** SDS-PAGE analysis of ZIKV NS2B/NS3$^{pro}$ solubility test. M: Protein marker, P: cell pellet, SN: supernatant. **B:** SDS-PAGE analysis of ZIKV NS2B/NS3$^{pro}$ after NI-NTA purification. M: Protein marker, W1: washing step without imidazole, W2-W4: washing step with imidazole (10, 20, 40 mM), E1-E3: imidazole elution steps (80, 250, 500 mM). **C:** Chromatogram of size exclusion chromatography of ZIKV NS2B/

NS3$^{pro}$. **D:** SDS-PAGE of ZIKV NS2B/NS3$^{pro}$ after size exclusion chromatography.
(TIF)

**S2 Fig. Preparation of CHIKV nsP2$^{pro}$.** Expression, purification and CD spectrum after the preparation process of CHIKV nsP2$^{pro}$. The CHIKV nsP2$^{pro}$ construct consists of 346 amino acids with a molecular weight of 39.38 kDa. The protein presented a single band on a denaturing SDS-PAGE gel with an apparent molecular mass of approximately 40 kDa. **A:** SDS-PAGE analysis of CHIKV nsP2$^{pro}$ solubility test. M: Protein marker, P: cell pellet, SN: supernatant. **B:** SDS-PAGE analysis of CHIKV nsP2$^{pro}$ after NI-NTA purification. M: Protein marker, W1: washing step without imidazole, W2-W3: washing step with imidazole (10, 40 mM), E1-E2: imidazole elution steps (250, 500 mM). **C:** Chromatogram of size exclusion chromatography of CHIKV nsP2$^{pro}$. **D:** SDS-PAGE of CHIKV nsP2$^{pro}$ after size exclusion chromatography.
(TIF)

**S3 Fig. Citrus plant flavonoids with inhibitory activity against ZIKV NS2B/NS3$^{pro}$ and CHIKV nsP2$^{pro}$.** Dose response curve for (A and B) HST and (C) HSD. Half maximum inhibitory concentration (IC$_{50}$) values were determined by nonlinear regression using 20 μM substrate (ZIKV NS2B/NS3$^{pro}$), 3 μM substrate (CHIKV nsP2$^{pro}$), 3 nM ZIKV NS2B/NS3$^{pro}$, 1 μM CHIKV nsP2$^{pro}$, with varying concentrations of the inhibitors. Data shown are the means ± SD from three independent measurements (n = 3). S1 Data contain the underlying data for the IC$_{50}$ value determination.
(TIF)

**S4 Fig. Fluorescence spectroscopy of Trp at 295 nm of ZIKV NS2B/NS3$^{pro}$ and CHIKV nsP2$^{pro}$ in the presence ligands.** HST and HSD titration experiments. Data shown are the means ± SD from three independent measurements (n = 3). **A:** Fluorescence of ZIKV NS2B/NS3$^{pro}$ under influence of HST titration demonstrated a red excitation shift of visible Trp (*). **B:** Binding saturation curve and modified Hill equation determined a K$_D$ value of 17.8 ± 2.9 μM for the ZIKV NS2B/NS3$^{pro}$-HST interaction. **C:** Fluorescence of CHIKV nsP2$^{pro}$ under influence of HSD titration. **D:** Binding saturation curve and modified Hill equation determined a K$_D$ value of 40.7 ± 2.0 μM for the CHIKV nsP2$^{pro}$-HSD interaction. **E:** Fluorescence of CHIKV nsP2$^{pro}$ under influence of HST titration demonstrated a red excitation shift of visible Trp (*). **F:** Binding saturation curve and modified Hill equation determined a K$_D$ value of 31.6 ± 2.5 μM for the CHIKV nsP2$^{pro}$-HST interaction.
(TIF)

**S5 Fig. K$_D$ determination using a modified Hill equation.** Based on fluorescence spectroscopy of Trp at 295 nm of ZIKV NS2B/NS3$^{pro}$ and CHIKV nsP2$^{pro}$ in the presence ligands. Intersection with x-axis corresponds to the logarithmic value of the K$_D$. **A:** ZIKV NS2B/NS3$^{pro}$-HST interaction. **B:** CHIKV nsP2$^{pro}$-HSD interaction. **C:** CHIKV nsP2$^{pro}$-HST interaction.
(TIF)

**S6 Fig. Thermal denaturation of ZIKV NS2B/NS3$^{pro}$ and CHIKV nsP2$^{pro}$ using fluorescence spectroscopy.** Data shown are the means ± SD from three independent measurements (n = 3). **A:** Fluorescence spectra during thermal denaturation of ZIKV NS2B/NS3$^{pro}$. **B:** Plot of the native protein fraction (fN) and the unfolding protein fraction (fU), during thermal denaturation from 20 to 85˚C. With increasing temperature fN decrease and fU increase, on the intersection of both curves the melting temperature (T$_m$) of 43˚C was determined for ZIKV NS2B/NS3$^{pro}$. **C:** Fluorescence spectra during thermal denaturation of CHIKV nsP2$^{pro}$. **D:** Plot of the native protein fraction (fN) and the unfolding protein fraction (fU), during thermal

denaturation from 20 to 85˚C. The melting temperature ($T_m$) of 47˚C was determined for CHIKV nsP2$^{pro}$.
(TIF)

**S7 Fig. Thermal denaturation of ZIKV NS2B/NS3$^{pro}$ and CHIKV nsP2$^{pro}$-HST and -HSD complexes using fluorescence spectroscopy.** Data shown are the means ± SD from three independent measurements (n = 3). **A:** Fluorescence spectra during thermal denaturation of ZIKV NS2B/NS3$^{pro}$-HST complex. **B:** Plot of the native protein fraction (fN) and the unfolding protein fraction (fU), during thermal denaturation from 20 to 85˚C. The melting temperature ($T_m$) of 43˚C was determined for ZIKV NS2B/NS3$^{pro}$ and for the ZIKV NS2B/NS3$^{pro}$-HST complex the $T_m$ increased to 49˚C. **C:** Fluorescence spectra during thermal denaturation of CHIKV nsP2$^{pro}$-HST complex. **D:** Plot of the native protein fraction (fN) and the unfolding protein fraction (fU), during thermal denaturation from 20 to 85˚C. The melting temperature ($T_m$) of 47˚C was determined for CHIKV nsP2$^{pro}$ and for the CHIKV nsP2$^{pro}$-HST complex the $T_m$ increased to 55˚C. **E:** Fluorescence spectra during thermal denaturation of CHIKV nsP2$^{pro}$-HSD complex. **F:** Plot of the native protein fraction (fN) and the unfolding protein fraction (fU), during thermal denaturation from 20 to 85˚C. The melting temperature ($T_m$) of 43˚C was determined for CHIKV nsP2$^{pro}$ and for the CHIKV nsP2$^{pro}$-HSD complex the $T_m$ changed to 52˚C.
(TIF)

**S8 Fig. Time dependent modifications during MD simulation of ZIKV NS2B/NS3$^{pro}$.** RMSD, RMSF, RoG and surface area changes over 200 ns of two independent MD runs. ZIKV NS2B (blue) and ZIKV NS3$^{pro}$ (green). The ZIKV NS2B cofactor is a small loop and is very flexible, which explain the bigger RoG and smaller surface area compared with NS3$^{pro}$. **A:** MD run1, **B:** MD run2.
(TIF)

**S9 Fig. Time dependent modifications during MD simulation of CHIKV nsP2$^{pro}$.** RMSD, RMSF, RoG and surface area changes over 200 ns of two independent MD runs. CHIKV nsP2$^{pro}$ MD run1 (dark blue) and CHIKV nsP2$^{pro}$ MD run2 (green).
(TIF)

**S10 Fig. Time dependent modifications during MD simulation of ZIKV NS2B/NS3pro-HST.** RMSD, RMSF, RoG and surface area changes over 200 ns of two independent MD runs. ZIKV NS2B (blue), ZIKV NS3pro (green) and HST (black). RMSD as function of time. A: ZIKV NS2B/NS3pro-HST MD run1. B: ZIKV NS2B/NS3pro-HST MD run2.
(TIF)

**S11 Fig. Time dependent modifications during MD simulation of CHIKV nsP2$^{pro}$-HST.** RMSD, RMSF, RoG and surface area changes over 200 ns of two independent MD runs. RMSD as function of time. **A:** CHIKV nsP2$^{pro}$-HST MD run1, CHIKV nsP2$^{pro}$ (dark blue) and HST (red). **B:** CHIKV nsP2$^{pro}$-HST MD run2, CHIKV nsP2$^{pro}$ (green) and HST (pink).
(TIF)

**S12 Fig. Time dependent modifications during MD simulation of CHIKV nsP2$^{pro}$-HSD.** RMSD, RMSF, RoG and surface area changes over 200 ns of two independent MD runs. RMSD as function of time. **A:** CHIKV nsP2$^{pro}$-HSD MD run1, CHIKV nsP2$^{pro}$ (dark blue) and HSD (red). **B:** CHIKV nsP2$^{pro}$-HSD MD run2, CHIKV nsP2$^{pro}$ (green) and HSD (pink).
(TIF)

**S13 Fig. Structural model of ZIKV NS2B/NS3$^{pro}$ after MD simulations with and without HST, RMSF and secondary structure changes.** ZIKV NS2B/NS3$^{pro}$: NS3$^{pro}$ in rosé, NS2B in

cyan, ZIKV NS2B/NS3$^{pro}$-HST: NS3$^{pro}$ in gray, NS2B in blue and HST in yellow. Secondary structure changes highlighted in red. **A:** Overlay of the RMSF of ZIKV NS2B/NS3$^{pro}$ with and without HST. **B:** Secondary structure changes over 200 ns of ZIKV NS2B/NS3$^{pro}$ with and without HST. I-IV labels small changes in the secondary structure. **C:** Structural overlay of ZIKV NS2B/NS3$^{pro}$ with and without HST. Secondary structure changes highlighted in red and the number code. **Right panel:** Zoom view on the secondary structure changes.
(TIF)

**S14 Fig. Structural model of CHIKV nsP2$^{pro}$ after MD simulations with and without HST and HSD, RMSF and secondary structure changes.** CHIKV nsP2$^{pro}$: protease domain in violet, methyl transferase domain in cyan, CHIKV nsP2$^{pro}$HST/HSD: protease domain in gray, methyl transferase domain in green, HST/HSD in yellow. **A:** Overlay of the RMSF of CHIKV nsP2$^{pro}$ with and without HSD and HST. **B:** Secondary structure changes over 200 ns of CHIKV nsP2$^{pro}$ with and without HSD and HST. I-VI labels small changes in the secondary structure. **C:** Structural overlay of CHIKV nsP2$^{pro}$ with and without HSD and HST. Secondary structure changes highlighted in red and the number code. **Right panel:** Zoom view on the secondary structure changes.
(TIF)

**S15 Fig. Atom numbers of HST and HSD. A:** HST and **B:** HSD.
(TIF)

**S16 Fig. ZIKV NS2B/NS3$^{pro}$ and CHIKV nsP2$^{pro}$ allosteric sites.** Surface view of the protease structures and the ligands. **A:** ZIKV NS2B/NS3$^{pro}$-HST complex. NS2B is colored in blue, NS3$^{pro}$ in gray, HST in yellow, active site in green and allosteric site in red. **B:** CHIKV nsP2$^{pro}$-HST complex. Protease domain is colored in gray, methyltransferase domain in green, HST in yellow, active site in blue and allosteric site in red. **C:** CHIKV nsP2$^{pro}$-HSD complex. Protease domain is colored in gray, methyltransferase domain in green, HSD in yellow, active site in blue and allosteric site in red.
(TIF)

**S17 Fig. Plaque reduction assay to determine the antiviral effect of HST and HSD against ZIKV and CHIKV.** Data shown are the means ± SD from three independent measurements (n = 3). There were no antiviral activities of HST and HSD observable. Virus control (black) and two inhibitor concentrations (grey). **A:** ZIKV under HST influence. **B:** CHIKV under HST influence and **C:** CHIKV under HSD influence.
(TIF)

**S1 Table. Drugs or inhibitors affecting flavivirus and alphavirus proteases or other virus target proteins.**
(DOCX)

**S2 Table. Contributing amino acids in the Substrate-binding sites of ZIKV NS2B/NS3$^{pro}$ and CHIKV nsP2$^{pro}$.**
(DOCX)

**S1 Data.**
(XLSX)

## Acknowledgments

We would like to thank Dr. Nivedita Cukkemane to prove read the manuscript. Further, we would like to thank members of the Multiuser Center for Biomolecular Innovation (IBILCE,

São Jose do Rio Preto, Brazil) and the Institute of Biological Information Processing (Forschungszentrum Jülich, Germany), without their help this project would not have been feasible.

## Author Contributions

**Conceptualization:** Raphael J. Eberle, Monika A. Coronado.

**Formal analysis:** Raphael J. Eberle, Carolina C. Pacca, Mauricio L. Nogueira, Monika A. Coronado.

**Investigation:** Raphael J. Eberle, Danilo S. Olivier, Carolina C. Pacca, Clarita M. S. Avilla, Marcos S. Amaral, Monika A. Coronado.

**Methodology:** Raphael J. Eberle, Carolina C. Pacca, Clarita M. S. Avilla, Monika A. Coronado.

**Resources:** Mauricio L. Nogueira, Dieter Willbold, Raghuvir K. Arni.

**Software:** Danilo S. Olivier, Marcos S. Amaral.

**Supervision:** Monika A. Coronado.

**Validation:** Raphael J. Eberle, Danilo S. Olivier, Carolina C. Pacca, Marcos S. Amaral, Monika A. Coronado.

**Writing – original draft:** Raphael J. Eberle.

**Writing – review & editing:** Dieter Willbold, Raghuvir K. Arni, Monika A. Coronado.

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
