## [Decision Letter · Decision Letter 0]

12 Nov 2020

PONE-D-20-32447

In vitro study of Hesperetin and Hesperidin as inhibitors of Zika and Chikungunya virus proteases

PLOS ONE

Dear Dr. Eberle,

Thank you for submitting your manuscript to PLOS ONE. After careful consideration, we feel that it has merit but does not fully meet PLOS ONE’s publication criteria as it currently stands. Therefore, we invite you to submit a revised version of the manuscript that addresses the points raised during the review process. Please consider each of the points brought up by the reviewers, specifically concerning rigor and technical soundness.

We look forward to receiving your revised manuscript.

Kind regards,

Bryan Mounce, Ph.D.

Academic Editor

PLOS ONE

Journal Requirements:

Reviewers' comments:

Reviewer's Responses to Questions

**Comments to the Author**

1. Is the manuscript technically sound, and do the data support the conclusions?

Reviewer #1: Yes

Reviewer #2: Partly

2. Has the statistical analysis been performed appropriately and rigorously? 

Reviewer #1: No

Reviewer #2: No

3. Have the authors made all data underlying the findings in their manuscript fully available?

Reviewer #1: Yes

Reviewer #2: No

4. Is the manuscript presented in an intelligible fashion and written in standard English?

Reviewer #1: No

Reviewer #2: No

5. Review Comments to the Author

Reviewer #1: The study by Erbele et al., investigates the capacity of two natural plant-based compounds (HSD and HST) to inhibit the protease activity of ZIKV NS2B/NS3pro and CHIKV NSP2pro in vitro. The authors demonstrate inhibitory effects of both compounds against the CHIKV protease while only HST displays a modest inhibitory effect against the ZIKV protease. Through a series of dose response experiments, fluorescence spectroscopy experiments, thermal stability assays, as well as in-silico molecular docking analyses, the authors define the IC50 values of their compounds in their in vitro protease activity assays and propose the inhibitors to act non-competitively by binding allosterically to both viral proteases, thus inducing a conformational change which the authors propose alters protease activity. For the most part the biochemical data are convincing, but with all purely biochemical studies, the biological relevance of the findings is unclear. Although these compounds have been shown to possess antiviral activity against CHIKV, whether they possess antiviral activity against ZIKV is unknown. The authors should attempt to demonstrate the antiviral activity of their compounds to support their proposed usage of HSD and HST as therapeutics to treat ZIKV and CHIKV infection/co-infection. The manuscript would also benefit from the addition of a few specific controls. Specific points for the authors to address are below:

1. The authors should test the antiviral activity of HSD and HST against ZIKV infection in vitro. It would also be useful to show the antiviral activity of these compounds against CHIKV infection in the authors hands but this is less critical as already shown by others.

2. It seems as if the binding affinities of HST and HSD for CHIKV and ZIKV proteases do not correlate with their inhibitor capacity against protease activity. Could the other comment on this observation?

3. Since the authors propose usage of HSD/HST or modified compounds, to treat ZIKV and CHIKV infected patients, some discussion on the potential maximal tolerated dose, pharmacological kinetics, and dose of administration should be added to the manuscript.

4. Do the authors hypothesize the anti-protease activity HSD and HST to be specific to these viral proteases or much broader? Since HST was previously shown to inhibit protease activity of snake venom, perhaps the activity is quite broad? The authors should discuss the potential for off-target effects of these compounds and how this would be tolerated by cells in vitro or human patients.

5. It would be useful for the authors to measure cytotoxicity levels of HSD and HST on relevant human cell in vitro using the compound doses tested in this paper.

6. Statistics should be added to figure 2.

Minor Comments:

1. Line 103/104: should be written to treat co-infections?

2. Some information about the strains the CHIKV and ZIKV recombinant proteins come from should be provided in the methods.

3. Figure 2: For ZIKV protease inhibition what is the % activity observed required to call a compound a hit. A positive and negative control compound would be helpful to interpret the results.

Reviewer #2: This study reports the inhibition of two viral proteases by two related flavonoids, HST and HSD, and proposes a model for how the compounds may be binding to allosteric site as inhibition is non-competitive. Considering no drugs exist to treat Zika and chikungunya virus infections, finding compounds that inhibit their viral proteases is valuable. Many flavonoids have already been reported to have antiviral effects, including the ones reported here. The current work shows that only HST inhibits Zika while both HST and HSD inhibit CHIKV protease. No other flavonoid (some listed in Table 1) was tested. The binding affinity was assessed by determining IC50, Ki, Kd and shift in melting temperature (DeltaTm). These measurements are not consistent with each other and there is no attempt in the manuscript to explain the underlying reasons (for example, IC50/Kd are similar for the two compounds against CHIKV but KD and deltaTm differ; Ki against ZIKV relatively low but not Kd).

The work suffers from several technical issues and rigor, as detailed below. In addition, the method for docking is not described (lines 426-427). Why HST inhibits Zika protease but not HSD is not clear from the computational results presented. Below are detailed responses to the Journal's review questions:

1. Is the manuscript technically sound, and do the data support the conclusions?

There are several major issues with replication, controls and technical quality.

(1) For the molecular dynamics simulations, replicates are missing to show reproducibility and reliability. In addition, RMSF and RMSD plots (supplementary figure 6) suggest potential problems with the simulations as the RMSDs look erratic and RMSFs are too high.

(2) Replicates for determining IC50 values (Figure S1) are not shown and the curves do not reach saturation, decreasing the reliability of the values calculated.

(3) Binding of inhibitor to the proposed site is not confirmed by either experimental mutagenesis or other methods.

(4) Number of experimental replicates (and associated error bars) are not stated in most cases.

(5) The source and purity of inhibitors tested are not provided.

(6) The protein purity and molecular weight is not confirmed.

(7) Description of equation (4) in Methods is unclear and inconsistent as some terms are defined as “percentage” while others as “amount” or “concentration” and how fluorescence units are converted to concentration (without measuring minimum or background) is not clear. (8) The reasoning behind fitting the same data to both equation (4) and (5) is not justified, and the parameters (m, n) from the fit are not reported.

(9) Using equation (6) requires justification or assumption of cooperative or 1-step unfolding.

(10) Control simulations with no-ligand bound protease are needed as changes seen (for example in secondary structure content) may be due to crystal contacts.

(11) Line 459: the conclusion that allosteric binding would change catalytic site conformation cannot be drawn from the results as there is no data to support this

(12) Relation between Ki and IC50 depends on the inhibition mechanism (among other things) and Equation 2 is for competitive inhibition, so should not be used for noncompetitive inhibition here.

2. Has the statistical analysis been performed appropriately and rigorously?

Major: Statistical rigor is lacking especially for reproducibility of MD results, and error analysis for Kd and IC50 determination as well as fluorescence measurements. The very short section on Statistical analysis says P values less than 0.05 were considered significant.

Minor: Significance of reported changes in secondary structure elements (lines 438) is not known. Error analysis and comparison with apo controls are needed to be able to draw these conclusions.

3. Have the authors made all data underlying the findings in their manuscript fully available?

The data points behind experimental measurement results are not provided.

4. Is the manuscript presented in an intelligible fashion and written in standard English?

The manuscript would benefit from copyediting to improve readability.

6. PLOS authors have the option to publish the peer review history of their article (what does this mean?). If published, this will include your full peer review and any attached files.

Reviewer #1: **Yes: **Scott B Biering.

Reviewer #2: No

---

## [Author Response · Author response to Decision Letter 0]

13 Jan 2021

Reviewer 1

Comment 1

The authors should test the antiviral activity of HSD and HST against ZIKV infection in vitro. It would also be useful to show the antiviral activity of these compounds against CHIKV infection in the authors hands but this is less critical as already shown by others.

Response 1

As suggested by the reviewer we have tested the molecules against ZIKV and CHIKV infection in vitro. The CHIKV results corroborate to the already published Ahmadi et al. 2016 (https://doi.org/10.1039/C6RA16640G). In Vero cells infected with ZIKV or CHIKV the compounds were practically inactive (see page 25, lines 666-667 and S17 Fig). This suggests that some modifications, as including a hydrophobic group in the molecule structure should be necessary to cross the cellular membrane and even hydrophobic moiety might be advantageous.

To analyze the action of the molecules against the viruses in infected cells it would be necessary to use ZIKV/CHIKV replicons. Unfortunately, we could not find a collaborator partner able to perform that experiment in this moment of the coronavirus crises. However, we are still looking for partners that can perform inhibition tests on virus replicons, especially for ZIKV, as Ahmadi et al. have already demonstrated the effect against CHIKV.

It is already very well characterized that the identification of a lead compound is the first step in making a new drug to treat a disease and we identified, by in vitro experiments, molecules that inhibit two different virus protease families. The molecules possess good binding affinities and low cytotoxicity. 

We are introducing both molecules as lead compounds and there is clearly space for improvement of the compound properties in vitro and in vivo by chemical modifications. The development processes take time and we are presenting here the effect of both molecules in enzymatic assays.

Comment 2

It seems as if the binding affinities of HST and HSD for CHIKV and ZIKV proteases do not correlate with their inhibitor capacity against protease activity. Could the other comment on this observation?

Response 2

Actually, we did a mistake in the analyses, using a wrong equation, which is to determine the Ki values (competitive inhibitors); however, we identified noncompetitive inhibitors. We corrected the manuscript appropriately.

Comment 3

Since the authors propose usage of HSD/HST or modified compounds, to treat ZIKV and CHIKV infected patients, some discussion on the potential maximal tolerated dose, pharmacological kinetics, and dose of administration should be added to the manuscript.

Response 3

Thank you for the suggestion, we included a paragraph in the results and discussion section “Pharmacokinetic view on the eligibility of HST and HSD as lead molecules targeting viral proteases” and table 5 that include pharmacological kinetics for HST and HSD, page 24 lines 629 to 644.

Comment 4

Do the authors hypothesize the anti-protease activity HSD and HST to be specific to these viral proteases or much broader? Since HST was previously shown to inhibit protease activity of snake venom, perhaps the activity is quite broad? The authors should discuss the potential for off-target effects of these compounds and how this would be tolerated by cells in vitro or human patients.

Response 4

Shawar et al. 2013, described the inhibitory effect of HST and HSD against trypsin. However, the determined IC50 values 104 µM (HST) and 127 µM (HSD) are 10 to 50 times higher than the concentrations we observed for our target proteins. The same applies to the human pancreatic elastase and HST (Page 25, lines 656 to 661). The text was changed appropriately. 

Regarding the question “If cells in vitro or human patients would tolerate these compounds”. We included in table 5 the maximal tolerated dose and in the literature, several studies showed that citrus plant flavonoids are safe, e.g.:

• Manach C, Morand C, Gil-Izquierdo A, Bouteloup-Demange C, Remesy C. Bioavailability in humans of the flavanones hesperidin and narirutin after the ingestion of two doses of orange juice. Eur. J. Clin. Nutr. 2003;57:235-242. https://doi.org/10.1038/sj.ejcn.1601547

• Srirangam R, Hippalgaonkar K, Avula B, Khan IA, Majumdar S. Evaluation of the intravenous and topical routes for ocular delivery of hesperidin and hesperetin. J. Ocul. Pharmacol. Ther. 2012;28:618-627. https://doi.org/10.1089/jop.2012.0040

• Maiti K, Mukherjee K, Murugan V, Saha BP, Mukherjee PK. Exploring the effect of hesperetin–HSPC complex—a novel drug delivery system on the in vitro release, therapeutic efficacy and pharmacokinetics. AAPS PharmSciTech. 2009;10:943. https://doi.org/10.1208/s12249-009-9282-6

Additionally, MTT assays using Vero cells with HSD and HST were performed and both compounds demonstrated a good tolerability page 25 lines 652 to 655.

Comment 5

It would be useful for the authors to measure cytotoxicity levels of HSD and HST on relevant human cell in vitro using the compound doses tested in this paper.

Response 5

Thank you for the comment, MTT assays were performed using Vero cells with HSD and HST. The results are shown in Fig. 8, pages 24 and 25 lines 645 to 655.

Comment 6

Statistics should be added to figure 2.

Response 6

We changed accordingly the figure 2.

Minor Comments:

Comment 1

Line 103/104: should be written to treat co-infections?

Response 1

We changed the sentence “Currently no appropriate antiviral treatment for CHIKV and/or ZIKV infection is available, neither to treat co-infections caused by these viruses” to “Currently, neither any appropriate antiviral treatment for CHIKV and/or ZIKV infection is available, nor for treatment of co-infections caused by these viruses”.

Comment 2

Some information about the strains the CHIKV and ZIKV recombinant proteins come from should be provided in the methods.

Response 2

We included the missing information for ZIKV (Page 5 lines 137 to 138) and CHIKV (Page 6 lines 144 to 145) strains.

Comment 3

For ZIKV protease inhibition what is the % activity observed required to call a compound a hit. A positive and negative control compound would be helpful to interpret the results.

Response 3

Figure 2 shows the results of an inhibition activity experiment used to evaluate the minimum inhibition of the proteases in a settled concentration of the molecules (in our case we have used 20 µM). In the first screening (data not shown), molecules that showed inhibition greater than 25 % were selected for subsequent experiments, for example, IC50. 

We used then the determined IC50 values to compare them with other inhibitor molecules already described in the literature, where we demonstrate the potential of both studied compounds and the ligand efficiency for them (Table 1). We are the opinion that the combination of inhibition, affinity and low cytotoxicity make both molecules interesting to be used as lead molecules.

Reviewer 2

Comment 1

For the molecular dynamics simulations, replicates are missing to show reproducibility and reliability. In addition, RMSF and RMSD plots (supplementary figure 6) suggest potential problems with the simulations as the RMSDs look erratic and RMSFs are too high.

Response 1

RMSF and RMSD plots in supplementary figures were corrected accordingly.

Replicates of ZIKV-HST, CHIKV-HST and CHIKV-HSD complexes were performed. The general information of MD simulation (duplicates) are included in the supplementary figures 10 to 12. Additionally, structural overlays of the corresponding ligand binding areas are included in figures 5 and 6 (pages 21 and 22). For ZIKV-HST, the simulation reproduces the position of HST and the involved amino acids (RMSD between both representative structures: 1.475 Å). In CHIKV-HST, the position of HST changes slightly between both simulations, but stay in the same pocket. Lys89 form a hydrogen bond in both simulations (RMSD between both representative structures: 1.259 Å). In CHIKV-HSD complex, the position of HSD also changes slightly between both MD simulations, nevertheless, the molecules kept in the same pocket site and not altering the amino acids that interact with the molecule, except for Met240 that, in run2, is not forming any more a hydrogen bond with the ligand. The other amino acids, which are involved in hydrogen bonds, Glu48, Val75 and Tyr77 are the matching between simulations. 

The duplicate simulations validate the binding of the molecules, and demonstrate the high probability that HST or HSD might interact in the proposed regions.

The running time for the MD simulations was 200 ns, which is longer when compared with several publications in the same field (examples, see below):

200 ns 

https://doi.org/10.1038/s41598-019-42935-y

150 ns 

https://doi.org/10.1080/07391102.2015.1046934

https://doi.org/10.1016/j.jmgm.2017.03.002

100 ns

https://doi.org/10.1016/j.lfs.2020.118080

50 ns

https://doi.org/10.1080/07391102.2020.1808077

We assume that the experimental time (200 ns) in combination with the replicates demonstrate the reproducibility and reliability of our results. 

Comment 2

Replicates for determining IC50 values (Figure S1) are not shown and the curves do not reach saturation, decreasing the reliability of the values calculated.

Response 2

The IC50 determination are based on triplicates (which was also already stated in the material and methods section). The IC50 curves in Fig S3 are based on mean value of the three replicates ± SD. A supplementary file with the experimental data was added accordingly.

Comment 3

Binding of inhibitor to the proposed site is not confirmed by either experimental mutagenesis or other methods.

Response 3

Yes, that is true. However, that was not our scientific question. We would like to identify inhibitor molecules from natural source that would be able to inhibit the protease and works as a lead molecule. By inhibition assays, inhibition mode and in silico binding interaction studies we were able to characterize and describe the inhibition mode of two citrus flavonoids. We suggested possible binding regions using MD simulations. We make clear in the manuscript that the binding regions we are showing have a potential to bind the ligands, but experimental methods are necessary to confirm the suggestion.

Comment 4

Number of experimental replicates (and associated error bars) are not stated in most cases.

Response 4

Actually, the number of experimental replicates were stated in the material and methods section. However, we included accordingly the number of experimental replicates and associated error bars in the figures, figure descriptions and main text.

Comment 5

The source and purity of inhibitors tested are not provided.

Response 5

We added the source and purity of the commercially purchased compounds (Page 7 lines 172 to 173).

Comment 6

The protein purity and molecular weight is not confirmed.

Response 6

We changed accordingly and included in the Results and Discussion section “Preparation of ZIKV NS2B/NS3pro and CHIKV nsP2pro” (Page 14 lines 364 to 368). Additionally figures were included in the supplementary material, S1 and S2 Figs.

Comment 7

Description of equation (4) in Methods is unclear and inconsistent as some terms are defined as “percentage” while others as “amount” or “concentration” and how fluorescence units are converted to concentration (without measuring minimum or background) is not clear. 

Response 7

The description of equation (4) was corrected accordingly. Actually, we measured minimum or background, as described in the material and methods section “All spectra were corrected for background intensities by subtracting the spectra of pure solvent measured under identical conditions”.

Comment 8

The reasoning behind fitting the same data to both equation (4) and (5) is not justified, and the parameters (m, n) from the fit are not reported.

Response 8

We included the missing information accordingly.

Comment 9

Using equation (6) requires justification or assumption of cooperative or 1-step unfolding.

Response 9

We assumed a 1-step unfolding, described in the material and method section, page 10, lines 267 to 268.

Comment 10

Control simulations with no-ligand bound protease are needed as changes seen (for example in secondary structure content) may be due to crystal contacts.

Response 10

Two independent MD simulations of CHIKV nsP2pro and ZIKV NS2B/NS3pro were performed. The general information of the MD simulations are shown in the supplementary figures S8 (ZIKV) and S9 (CHIKV). The RMSD of the ZIKV representative structures of run1 and run2 was 1.191. The RMSD of the CHIKV representative structures of run1 and run2 was 1.777. 

Comment 11

Line 459: the conclusion that allosteric binding would change catalytic site conformation cannot be drawn from the results as there is no data to support this.

Response 11

We suggested that the binding of HST to the possible ZIKV NS2B/NS3pro allosteric site influence the active site. We added the following sentence; “Additional experiments are needed to confirm this assumption.” (Page 21, line 538). 

Comment 12

Relation between Ki and IC50 depends on the inhibition mechanism (among other things) and Equation 2 is for competitive inhibition, so should not be used for noncompetitive inhibition here.

Response 12

Equation 2 and the corresponding Ki values were removed from the manuscript accordingly.

1. Has the statistical analysis been performed appropriately and rigorously?

Comment 1

Major: Statistical rigor is lacking especially for reproducibility of MD results, and error analysis for Kd and IC50 determination as well as fluorescence measurements. The very short section on Statistical analysis says P values less than 0.05 were considered significant.

Response 1

We corrected accordingly.

Comment 2

Minor: Significance of reported changes in secondary structure elements (lines 438) is not known. Error analysis and comparison with apo controls are needed to be able to draw these conclusions.

Response 2

We corrected accordingly.

Comment 3

Have the authors made all data underlying the findings in their manuscript fully available?

The data points behind experimental measurement results are not provided.

Response 3

We included additional Data in the supplementary material accordingly.

Comment 4

Is the manuscript presented in an intelligible fashion and written in standard English?

The manuscript would benefit from copyediting to improve readability.

Response 4

The manuscript was copyedited by Dr. Nivedita Cukkemane to improve the readability.

---

## [Editor Report · Decision Letter 1]

18 Jan 2021

In vitro study of Hesperetin and Hesperidin as inhibitors of Zika and Chikungunya virus proteases

PONE-D-20-32447R1

Dear Dr. Eberle,

We’re pleased to inform you that your manuscript has been judged scientifically suitable for publication and will be formally accepted for publication once it meets all outstanding technical requirements.

Kind regards,

Bryan Mounce, Ph.D.

Academic Editor

PLOS ONE

---

## [Editor Report · Acceptance letter]

19 Feb 2021

PONE-D-20-32447R1 

*In vitro* study of Hesperetin and Hesperidin as inhibitors of Zika and Chikungunya virus proteases 

Dear Dr. Eberle:

I'm pleased to inform you that your manuscript has been deemed suitable for publication in PLOS ONE. Congratulations! Your manuscript is now with our production department. 

Kind regards, 

on behalf of

Dr. Bryan Mounce 

Academic Editor

PLOS ONE